# Scale-Invariant Continuous Implicit Neural Representations For Remote Sensing Object Counting

## Abstract

Many object counting methods rely on density map estimation (DME) using convolutional neural networks (CNNs) on discrete grid image representations. However, these methods struggle with large variations in object size or input image resolution, typically due to different imaging conditions and perspective effects. Worse yet, discrete grid representations of density maps result in information loss with blurred or vanished details for low-resolution inputs. To overcome these limitations, we design Scale-Invariant Implicit neural representations for counting (SI-INR) to map arbitrary-scale input signals into a continuous function space, where each function produces density values over continuous spatial coordinates. SI-INR achieves robust counting performances with respect to changing object sizes, extensive experiments on commonly used diverse datasets have validated the proposed method.

## 1 Introduction

Understanding the distribution and abundance of people as well as geographic entities such as buildings and cars within these environments becomes crucial for various "smart city" applications, such as urban planning, traffic management and beyond. Object counting holds promising potential for such tasks, which have also been studied in other domains too, including crowd counting for security (Ma et al., 2019; Li et al., 2018), animal crowd estimations (Ma et al., 2015), and cell counting for biomedicine (Paul Cohen et al., 2017). Successful counting methods have been developed by introducing deep learning (Fu et al., 2015; Wen et al., 2021) and self-attention (Gao et al., 2020; Lin et al., 2022). In recent years, the best-performing methods are mostly based on density map estimation (DME) (Ma et al., 2019; Gao et al., 2022; Lin et al., 2022; Wan et al., 2021), training convolutional neural networks (CNNs) to generate discrete density maps.

However, several challenges persist in applying current deep learning methods for reliable counting: 1) Scale-Dependence: CNNs (LeCun et al., 1998) lack intrinsic scale equivariance, leading to degraded performance when object sizes deviate from those seen during training. This issue is particularly pronounced for inputs at resolutions differing from the training set, as CNNs rely on fixed receptive fields that cannot dynamically adapt to scale variations; 2) Expressiveness-Bottleneck: Traditional grid-based density maps approximate object distributions with Gaussian kernels, imposing a fixed spatial structure that misaligns with irregular object arrangements (Wan & Chan, 2019). Gaussian smoothing reduces noise but blurs local details, degrading fidelity in dense and sparse regions, and limiting counting accuracy in complex scenes.

To address these issues, we design a new object counting framework named Scale-Invariant Implicit Neural Representations (SI-INR) mapping arbitrary-scale discrete images into 2D continuous functions which are invariant to the object or structure scales. This allows the model to preserve the fine details and reduce potential information loss for better counting accuracy and generalizability. Moreover, the scale-invariance, an important property for the mapping between input images and output density maps, is explicitly introduced as the inductive bias of model itself to potentially improve data efficiency and model robustness. Our main contributions can be summarized as follows:

1. We propose Scale-Invariant Continuous Implicit Neural Representations (SI-INR), an object counting framework mapping discrete grid signals into continuous 2D functions which are invariant to image scaling.
2. As a proof of concept, we give a realization of SI-INR using existing Scale-Equivariant Steerable Network (SESN) (Sosnovik et al., 2019) and a novel deep neural operator based INR module. A sampling-based optimization objective is then derived for efficient model training.
3. We conduct extensive experiments to evaluate the effectiveness of our SI-INR on object counting, demonstrating notable advancements in performance, especially on the remote sensing counting dataset.

## 2 Related Work

Object counting: Object counting, e.g. well-studied crowd counting (Lin et al., 2001), has been mostly developed by detecting or segmenting individual objects in the scene. End-to-end learning to directly map image features to object counts has been the most successful counting strategy with rapid advancements in deep learning Krizhevsky et al. (2012); Wang et al. (2015), especially for object counting in densely populated scenes (Chan et al., 2008). Counting based on Density Map Estimation (DME) using convolutional neural networks (CNNs) (Lempitsky & Zisserman, 2010; Fu et al., 2015; Ranjan et al., 2018; Sindagi & Patel, 2019) to preserve translation-invariant multi-scale image features has shown superior performance over conventional object counting techniques. More recent ASPDNet (Gao et al., 2020) and PSGCNet (Gao et al., 2022) have integrated attention, deformable convolution, pyramidal scale modules (PSM) to address challenges in counting such as complex cluttered backgrounds, viewing perspective, object appearance, and size variability. Besides, Huang et al. (2023) propose an optimized global regression model EfreeNet which is more more annotation-efficient. Yi et al. (2023) introduce a lightweight multiscale context fusion module (LMCFM) and a lightweight counting scale pooling module (LCSPM) to reduce the numbers of network parameters and computing cost. These models have achieved state-of-the-art (SOTA) counting performance on the RSOC (Remote Sensing Object Counting) dataset (Gao et al., 2020).

Scale-equivariance and invariance: The concept of scale-equivariance and invariance was first proposed in image processing and computer vision (Lowe, 1999; 2004). To handle variations in scale effectively, multi-scale features can be learned by applying the convolutions to several rescaled versions of the images or feature maps in every layer (Kanazawa et al., 2014; Marcos et al., 2018) or by rescaling trainable filters (Xu et al., 2014). Cai et al. (2016) proposed a pyramidal structure to learn scale-dependent features, which is widely used in the object detection field nowadays. Later, Gaussian scale-space theory (Lindeberg, 1994) and group theory (Cohen & Welling, 2016) have been widely used for achieving scale-equivariance and invariance. Yang et al. (2023); Lindeberg (2022) parameterized convolutional filters as a linear combination of Gaussian derivative filters with different scales, and achieved scale-equivariance in image classification and segmentation tasks. Unlike models rooted in Gaussian scale-space theory, Sosnovik et al. (2019) proposed a Scale-Equivariant Steerable Network (SESN), which utilizes steerable filters parameterized by a trainable linear combination of pre-calculated Hermite basis functions. These models all first build a scale-equivariant model, and use a simple pooling layer or rescale the outputs to convert the model into a scale-invariant one if needed (Sosnovik et al., 2019). However, such methods have significant demands on memory and computational resources and can lose information when doing the equivariance to invariance conversion. In SI-INR, we adopt a scale-equivariant model to learn a deep representation that is equivariant to input scale variations, and introduce a scale-invariant model to convert the equivariant mapping to an invariant one.

Implicit neural representations: Implicit Neural Representations (INRs) allow for continuous flexible representations of complex objects and scenes in computer vision (Mescheder et al., 2019; Oechsle et al., 2019; Barrowclough et al., 2021; Wang et al., 2022). Together with recently developed positional encoding strategies (Tancik et al., 2020; Sitzmann et al., 2020) and end-to-end hypernetwork-based learning (Dupont et al., 2022; Lee et al., 2024; Kim

et al., 2023), efficient training to capture high-frequency details has been achieved to better model complex natural signals.

However, hypernetwork-based INR models are not translation-equivariant, making them unsuitable for handling scale variations in input signals. As a result, these models typically require the input to be of a fixed size. More recent Hierarchical Neural Operator Transformer (HiNOTE) (Luo et al., 2024) integrates neural operators in implicit neural representation, which can preserve more local information and improve the generalizability of INR models.

In SI-INR, we adopt a lightweight INR implementation and replace the traditional coordinate input by a continuous latent. In this case, our INR model transforms continuous latent to the continuous representation of targets, which can be seen as a deep neural operator (Kovachki et al., 2023). This approach improves stability, accelerates model training, and offers greater flexibility when incorporating images of varying sizes during training.

## 3 Method

We propose a novel object counting framework, Scale-Invariant Implicit Neural Representation (SI-INR), that explicitly models scale-invariance in the adopted continuous INR for robust object counting. We start the discussion by first presenting the problems of existing methods in Section 3.1. Next, we describe our solution and concept of SI-INR with the corresponding analysis in Section 3.2.1. We then describe the detailed construct of SI-INR using SESN and a deep neural operator based INR architectures in Section 3.2.2. Lastly in Section 3.3, we provide our sampling based model training for SI-INR on object counting.

### 3.1 Problem statement

In many real-world computer vision applications, input images can vary widely in dimension and size, which requires the corresponding object detection and counting models to be able to robustly identify objects ranging from a few pixels to thousands of pixels in scale. While existing off-the-shelf CNNs often take the input images of the fixed size, the flexibility to handle arbitrary resolution inputs and generate arbitrary resolution outputs is highly desirable to take the best advantage of heterogeneous data available.

For a given image $\mathbf{I}$, existing methods aim to establish a mapping $f : \mathbb{R}^{d_{\mathbf{I}}} \Rightarrow \mathbb{R}^{d_{\mathbf{D}^{gt}}}$ from input image $\mathbf{I} \in \mathbb{R}^{d_{\mathbf{I}}}$ to corresponding outputs $\mathbf{D}^{gt} \in \mathbb{R}^{d_{\mathbf{D}^{gt}}}$, where both input images and outputs are in typical discrete-grid representations and the outputs' resolution $d_{\mathbf{D}^{gt}}$ is typically lower than inputs' resolution reflected by the input dimension $d_{\mathbf{I}}$.

However, since traditional convolution models are not scale-invariant and always have a fixed downsampling ratio, these methods usually struggle with handling varying resolution inputs and objects in different scales. where $f(p_1(h)(\mathbf{I}_a)) \neq f(\mathbf{I}_a)$. Here $h$ denotes an element of the Scale-Translation group $H$ and represents one scale-translation operator, $p_1(\cdot)$ denotes the corresponding group actions of $h$ acting on the image domain. We provide a more detailed explanation of scale equivariance in Appendix A.1.

To address this limitation, we propose SI-INR to model the mapping for continuous signal representations. SI-INR learns a mapping $\Psi : \mathcal{I} \to \mathcal{F}$ from input image space $\mathcal{I}$ to the continuous function space $\mathcal{F}$.

$$\Psi(p_1(h)(\mathbf{I}_a))(\mathbf{x}) = \Psi(\mathbf{I}_a)(\mathbf{x}) = \mathbf{D}^{gt}(\mathbf{x}), \tag{1}$$

where we emphasize that the input image space $\mathcal{I}$ here is more flexible considering arbitrary normalized spatial coordinates, $\mathbf{x} \in [0,1]^2$, sampling from continuous image space. $\Psi(\mathbf{I}_a)$ denotes the predicted continuous representation of density maps for $\mathbf{I}_a$. $\mathbf{D}^{gt}$ denotes a continuous ground truth. Due to this formulation, SI-INR allows the learned model to handle varying resolution inputs, detect as many target objects at different scales as possible, and generate arbitrary resolution outputs.

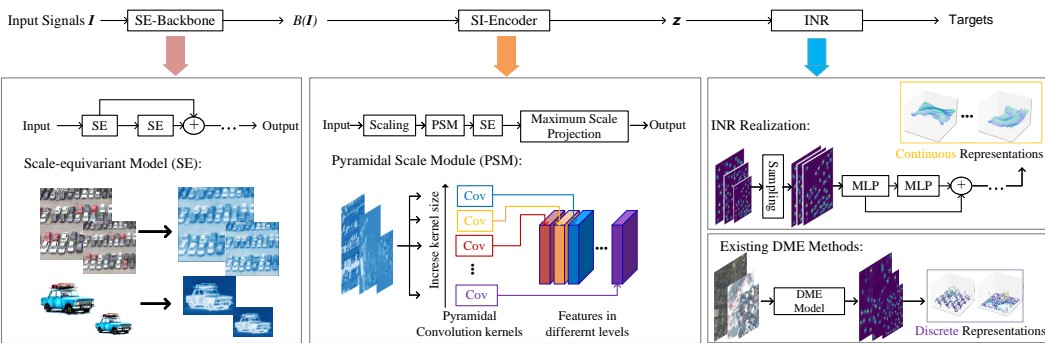

Figure 1: Schematic diagram of Scale-Invariant Implicit Neural Representation (SI-INR) and existing Density Map Estimation (DME) methods. SI-INR learns scale-invariant continuous representations in three steps: first, a scale-equivariant backbone is designed to extract deterministic scale-equivariant features; then, a scale-invariant encoder is adopted to aggregate scale-equivariant features in different scales; and finally, an INR decoder converts extracted features into an invariant output, a continuous representation of task targets. Visualization of continuous and discrete representations demonstrates that continuous representations preserve more information, leading to better reconstruction of the continuous output.

## 3.2 Scale-Invariant Implicit Neural Representations

Following the above formulation, to achieve a scale-invariant mapping from the image space $\mathcal{I}$ to the function space $\mathcal{F}$, we propose the SI-INR modular framework consisting of three components: a Scale-Equivariant backbone (SE-Backbone), a Hybrid Pyramidal Scale Module (HPSM), and an INR decoder. The SE backbone is designed to extract deterministic scale-equivariant features resilient to different resolutions of inputs and sizes of objects; then the HPSM merges scale-equivariant features in different scales; and finally, the INR decoder converts extracted features into a scale-invariant output, which is a continuous representation of density maps.

### 3.2.1 Model Components

Traditional computer vision backbones struggle with scale variations of objects where different sizes of the same objects can have different appearances in the feature maps. This results in inconsistent counting estimates and requires the model to have higher capacity to memorize the same objects in different scales (Zhan et al., 2022).

To construct a model which is data efficient and capable of handling unseen resolution inputs, we want our model to be scale-invariant so that consistent outputs can be generated with respect to varying scales of objects in input images. As the combination of scale-equivariant and scale-invariant models is scale-invariant (Sosnovik et al., 2019), we can construct the model by ensuring that either each component in the framework is invariant, or the framework appropriately combines scale-equivariant and scale-invariant components. It is more reasonable to implement a scale-equivariant mapping first to preserve fine details (Sosnovik et al., 2019), and then convert the equivariant mapping into a scale-invariant one for consequent prediction tasks.

SI-INR adopts this strategy and takes three steps to achieve scale-invariant mapping for arbitrary-scale signals as shown in Figure 1: first, a scale-equivariant model $B(\cdot)$ is adopted to extract deterministic features so that scale changes in objects will only affect the scale of feature maps while keeping the appearance:

$$B(p_1(h)(\mathrm{I}_a)) = p_B(h)(B)(\mathrm{I}_a), \tag{2}$$

where $p_B(\cdot)$ denotes the corresponding group actions of $h$ acting on the feature domain. We emphasize that any scale-equivariant method can be adopted into SI-INR to handle different tasks.

Second, a scale-invariant converter transforms the equivariant features into scale-invariant ones for consequent prediction tasks. At this step, a scale-invariant encoder $E(\cdot)$ is adopted to integrate extracted equivariant features:

$$E(p_B(h)(B(\mathrm{I}_a))) = E(B(\mathrm{I}_a)). \tag{3}$$

Equation (3) demonstrates that the output of $E(\cdot)$ is invariant to scale changes in the input signals. While $E(\cdot)$ can map the same signal at different scales into one latent space, it cannot map different signals at varying scales into a unified space. To address this problem, we introduce a scale-invariant continuous representation mapping $\mathcal{H}$. For any query image $\mathrm{I}_a$, $\mathcal{H}$ maps the output of $E(\cdot)$ into a continuous function $u_a : [0,1]^2 \to \mathbb{R}_{\geq 0}$, which means the corresponding density value for arbitrary normalized query position $\mathbf{x}$ can be predicted by evaluating the mapped continues function at $\mathbf{x}$: $u_a(\mathbf{x})$. In SI-INR, we extend the function to $u(x; z, \theta_{\mathrm{INR}})$, where $z \in \mathbb{R}^m$ represents the latent features extracted by the encoder. Thus, SI-INR learns a conditional continuous representation of the input by optimizing over $\theta_{\mathrm{INR}}$ and $z$. This allows for task-specific predictions such as continuous density estimation.

Training a continuous representation learner not only achieves a uniform format of output for different scales input but also enables flexible training sample-based algorithms to enhance the training as we discuss in Section 3.3.

We now prove the scale-invariance of our SI-INR for any scale-translation action on $\mathrm{I}_a$ in Theorem 1.

**Theorem 1** Given a scale-translation operation $h$ and an input image $\mathrm{I}_a$, SI-INR is scale-invariant:

Proof:

$$\Psi(p_1(h)(\mathrm{I}_a))(\mathbf{x}) = \mathcal{H}(E(B(p_1(h)(\mathrm{I}_a)))(\mathbf{x}) = \mathcal{H}(E(B(\mathrm{I}_a)))(\mathbf{x}) = \Psi(\mathrm{I}_a)(\mathbf{x}), \tag{4}$$

SI-INR not only is invariant to the change of scales, but also maps signals into one latent continuous function space.

### 3.2.2 SESN and Deep neural operator based Realization

In order to achieve the scale-invariant model, we first map the input signals into an equivariant space through a scale-equivariant backbone $B(\cdot)$. Thanks to the efficient INR-based equivariant-to-invariant conversion architecture, any scale-equivariant models can be applied in SI-INR, we choose SESN (Sosnovik et al., 2019) to build our scale-equivariant backbone $B(\cdot)$ as an example. The introduction of SESN is given in Appendix A.2. Since the summation of two scale-equivariant models is still scale-equivariant by Lemma 1 in Appendix A.2, we build our backbone based on the residual architecture, where the input is added to the output of each equivariant layer.

In SI-INR, the scale-invariant converter encoder $E(\cdot)$ is designed by a combination of a scaling operator $S(\cdot)$ and a CNN-based model $G$:

$$\begin{aligned}
E(B(p_1(h_1)(\mathbf{I}_a))) &= G(S(B(p_1(h_1)(\mathbf{I}_a)))) = G(p_B(h' \cdot h_1^{-1}) p_B(h_1)(B(\mathbf{I}_a))) \\
&= G(p_B(h')(B(\mathbf{I}_a))) = E(B((\mathbf{I}_a))),
\end{aligned} \tag{5}$$

where the scaling operator rescales any feature map $p_B(h_1)(B(\mathbf{I}_a))$ into a consistent scale: $p_B(h')(B(\mathbf{I}_a))$, $S(\cdot) = p_B(h' \cdot h_1^{-1})$. This scaling operator ensures the invariant property of our encoder $E(\cdot)$. Note that $h'$ is a hyperparameter which can rescale the derived equivariant features into a more reasonable size $\mathbb{R}^{l_w \times l_h \times C_S^{\mathrm{L}}}$, where $l_w$, $l_h$, and $C_S^{\mathrm{L}}$ denote the size and number of channels.

Prior knowledge about the original scales of test images can further improve the prediction accuracy of SI-INR. In the cases where the images in the dataset are unscaled and mutually independent, we can maintain the scale-invariance property by setting the scaling factor to 1.

In the CNN-based model $G$, we propose a Hybrid Pyramidal Scale Module (HPSM) consisting of a Pyramidal Scale Module (Gao et al., 2022) that implements convolutions with increasing kernel size to detect equivariant features in different scales, and scale-invariant convolutions (SESC with maximum scale projection) to convert extracted features into invariant ones.

Finally, SI-INR efficiently utilizes extracted information by introducing an implicit neural representation model $\mathcal{H}$ which maps different signals into a specific continuous function space. This mapping fully utilizes its continuous property to predict and recover fine details. Representing these deterministic features in a continuous function can capture more information compared with a discrete one. In our experiments in Section 4, SI-INR achieves better counting performance and we show that increasing the number of samples increases counting accuracy for small target objects such as compact cars.

$\mathcal{H}$ outputs a continuous representation $u_a$ of the target output for image $\mathbf{I}_a$, Here we give the expression of the INR decoder:

$$\mathcal{H}(E(B(\mathrm{I}_a)), \boldsymbol{\theta}_{INR})(\mathbf{x}) = u_a(\mathbf{x}|\mathbf{z}^a, \boldsymbol{\theta}_{INR}), \tag{6}$$

where $\boldsymbol{\theta}_{INR}$ denotes the trainable parameters and the INR model consists of $L_{INR}$ linear layers, $\boldsymbol{\theta}_{INR} = [\boldsymbol{W}_1, \boldsymbol{b}_1 \ldots, \boldsymbol{W}_{L_{INR}}, \boldsymbol{b}_{L_{INR}}]$.

To build a more flexible INR model that can handle complex scenarios, we generate a continuous latent $\mathbf{z}^a$, for any query position x, we compute corresponding $\mathbf{z}_{\mathrm{x}}^a$ by sampling from $\mathbf{z}^a$ and set it as the INR model's input.

In this way, our INR model treats input features as continuous functions instead of 2D discrete arrays, which can fully utilize local details in downstream analyses (Luo et al., 2024). We call this continuous-to-continuous INR module a deep neural operator based INR.

With all these, the predicted value at position x can be estimated as:

$$u_a(\mathbf{x}) = \boldsymbol{\phi}_{L_{INR}}^{INR}(\boldsymbol{W}_{L_{INR}}^T \ldots \boldsymbol{\phi}_1^{INR}(\boldsymbol{W}_1^T(\mathbf{z}_{\mathrm{x}}^a) + \boldsymbol{b}_1) \cdots + \boldsymbol{b}_{L_{INR}}), \tag{7}$$

where $\boldsymbol{\phi}^{INR}(\cdot)$ denotes the activation function.

We now prove the scale-invariance of our SI-INR's realization. Given a scale-translation operation $h$ and an input image $\mathrm{I}_a$, we have:

$$\begin{aligned} u_{p_{\mathrm{I}_a}(h)(\mathrm{I}_a)}(\cdot) &= \mathcal{H}(E(p_B(h' \cdot h^{-1})(B(p_{\mathrm{I}_a}(h)(\mathrm{I}_a)))), \boldsymbol{\theta}_{INR})(\cdot) \\ &= \mathcal{H}(E(p_B(h' \cdot h^{-1})(p_B(h)(B(\mathrm{I}_a)))), \boldsymbol{\theta}_{INR})(\cdot) \\ &= \mathcal{H}(E(p_B(h')(B(\mathrm{I}_a))), \boldsymbol{\theta}_{INR})(\cdot) = u_a(\cdot), \end{aligned} \tag{8}$$

where $p_{\mathrm{I}_a}(h)$ and $p_B(h)$ denote the group actions of $h$ on the input image domain and $B(\cdot)$'s output domain. For any scale-translation action $p_{\mathrm{I}_a}(h)$, the output for image $\mathrm{I}_a$ is always $u_a(\cdot)$.

### 3.3 Training with Regional Sampling

To train a continuous representation of the density map, a continuous ground truth is needed. We achieve this by directly constructing the likelihood function of any position $\mathbf{x}$ given label $y_n$ as $p(\mathbf{x} \mid y_n) = \mathcal{N}(\mathbf{x}; \mathbf{m}_n, \sigma^2 \mathbf{1}_{2\times2})$. The density map $\mathbf{D}^{gt}$ is then modeled as 2D stochastic processes in the continuous spatial domain:

$$\mathbf{D}^{gt}(\mathbf{x}) \stackrel{\text{def}}{=} \sum_{n=1}^{N} \mathcal{N}(\mathbf{x}; \mathbf{m}_n, \sigma^2 \mathbf{1}_{2\times2}) = \sum_{n=1}^{N} \frac{1}{\sqrt{2\pi}\sigma} \exp\left(-\frac{\|\mathbf{x} - \mathbf{m}_n\|_2^2}{2\sigma^2}\right), \tag{9}$$

where $\mathbf{x}$ denotes the normalized spatial coordinates, $\mathbf{x} \in [0, 1]^2$, $\mathcal{N}(\mathbf{x}; \mathbf{m}_n, \sigma^2 \mathbf{1}_{2\times2})$ denotes the 2D Gaussian distribution with the mean $\mathbf{m}_n$ and isotropic covariance matrix $\sigma^2 \mathbf{1}_{2\times2}$. Similar to Ma et al. (2019), we incorporate a Bayesian counting loss function in SI-INR, which is robust to noise and object occlusion. Together with the MAE counting loss, the

final minimization objective is:

$$\mathcal{L} = \frac{1}{A} \sum_{a=1}^{A} \Big[ \sum_{n=0}^{N} (\mathbb{E}_{\mathbf{x} \sim p_s(\mathbf{x})}(c_{n,a}) - \mathbb{E}_{\mathbf{x} \sim p_s(\mathbf{x})}(c_{n,a}^{\mathrm{gt}})) + \kappa (\sum_{n=0}^{N} (\mathbb{E}_{\mathbf{x} \sim p_s(\mathbf{x})}(c_{n,a})) - N) \Big], \quad (10)$$

where $c_{n,a}^{gt}(\cdot)$ and $c_{n,a}(\cdot)$ denote the contribution of $\mathbf{D}_a^{gt}$ to the $n$-th object label, $p_s(\mathbf{x})$ is any probability distribution of $\mathbf{x}$ which enables our model to be trained using any existing stochastic optimization algorithm, and $\kappa$ is a hyperparameter that balances the two loss function terms. To efficiently compute the loss, we introduce a regional sampling approach, where $\mathbf{x}$ is uniformly sampled from a pre-defined grid. The loss function is then rewritten as:

$$\mathcal{L} = \frac{1}{A} \sum_{a=1}^{A} \Big[ \sum_{n=0}^{N} (\mathbb{E}_{\mathbf{x} \sim \mathrm{Unif}[0,1]^2}(c_{n,a}) - \mathbb{E}_{\mathbf{x} \sim \mathrm{Unif}[0,1]^2}(c_{n,a}^{\mathrm{gt}})) + \kappa (\sum_{n=0}^{N} (\mathbb{E}_{\mathbf{x} \sim \mathrm{Unif}[0,1]^2}(c_{n,a})) - N) \Big].$$

$$(11)$$

The detailed derivation is provided in Appendix A.3. The count predictions can be hereby obtained by sampling uniformly over the normalized image domain and computing the summation of $\mathbf{D}_a(\mathbf{x})$.

## 4 Experiments

### 4.1 Experimental Setup

**Datasets.** We evaluate the model's performance on three challenging datasets: (1) the Remote Sensing Object Counting (RSOC) dataset (Gao et al., 2020); (2) the Car Parking Lot Dataset (CARPK) (Hsieh et al., 2017); and (3) the Pontifical Catholic University of Parana+ Dataset (PUCPR+) (Hsieh et al., 2017). Details on the datasets and the train-test split are provided in Appendix B.1.

**Baselines.** We compare our SI-INR with three baseline methods: ASPDNet (Gao et al., 2020), an attention-based network with scale pyramid and deformable convolutions; PSGC-Net (Gao et al., 2022), which integrates pyramidal scale and global context modules; and eFreeNet (Huang et al., 2023), an ensemble of first-rank-then-estimate networks. Network architecture details are available in Appendix B.2.

**Implementation.** We implement our scale-equivariant backbone by stacking four SESC layers (Sosnovik et al., 2019) with residual connections where the kernel size of SESC is set to 3 to reduce computational costs. Our scale-invariant encoder $E(\cdot)$ consists of two parts. The first part involves a scaling operator, the second part involves a VGG-19 network (Simonyan & Zisserman, 2014) to learn deep features, following a pyramidal scale module that has increasing kernel sizes from $3 \times 3$ to $11 \times 11$, and applies SESC layers with maximum scale projection. Our INR-based decoder network consists of four fully connected layers with residual connections, and one fully connected layer with learnable parameters to output raw density maps. We use the Adam (Kingma & Ba, 2014) optimizer for both our SI-INR and baseline models, and set the learning rate to be $1e-4$. We initialize parameters in SI-INR by random sampling from a Gaussian distribution $\mathcal{N}(0, 0.01^2)$. We set $\sigma = 8$ in our loss function and $S_{INR} = 64$ when generating density maps for different inputs unless specified. We evaluate our SI-INR with baseline models on the RSOC dataset, CARPK dataset, and PUCPR+ dataset, following Huang et al. (2023)'s setup with RSOC images resized into $512 \times 512$. Data augmentation has been implemented during model training by randomly flipping the input images horizontally. We select the models with the lowest RMSE and proper density maps in the first 300 training epochs and report the results. We run all our experiments with the fixed random seed 64 on a workstation with a NVIDIA V100 32GB GPU. We adopt two widely used metrics in object counting tasks following previous work (Gao et al., 2020; Ma et al., 2019) to evaluate baselines and our model: the Mean Absolute Error (MAE) and the Root Mean Squared Error (RMSE). We additionally compare SI-INR with SOTA crowd-counting methods on the UCF-QNRF dataset in Appendix B.5.

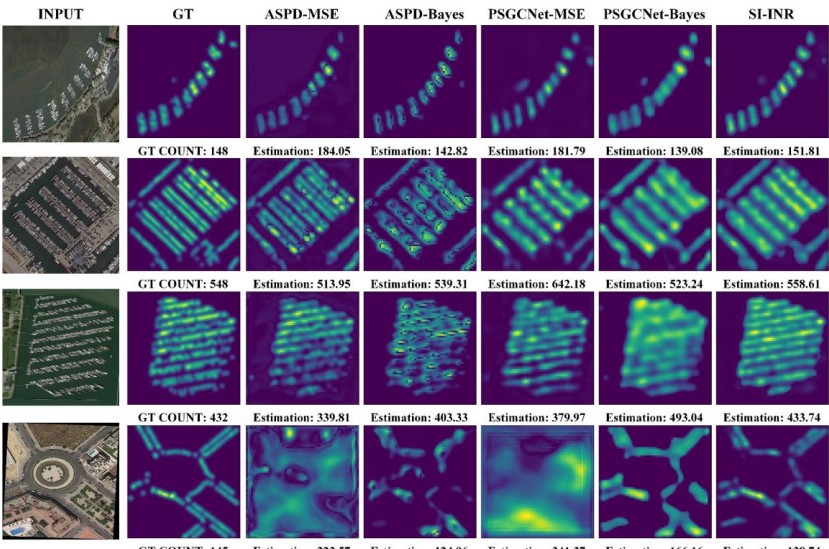

Figure 2: Predicted density maps by SI-INR and other baselines for four test images from RSOC. The test images (Test Images) and corresponding density maps (GT) are randomly sampled. The illustrated density maps are predicted by PSGCNet with MSE loss (PS-GCNet+MSE), PSGCNet with Bayesian counting loss (PSGCNet+Bayes), ASPDNet with MSE loss (ASPD+MSE), ASPDNet with Bayesian counting loss (ASPD+Bayes), and SI-INR. Warmer colors denote higher values while cooler colors denote lower values.

## 4.2 Main Results

**Quantitative Results.** Our performance evaluation on different benchmark datasets with the reported experimental results on RSOC in Table 1, and CARPK and PUCPR+ in Table 2, respectively. It shows hat SI-INR significantly improves the MAE on all the datasets compared with our baseline model PSGCNet. Also, SI-INR achieves comparable results, especially on PUCPR+ datasets. Note that the RSOC small-vehicle and RSOC ship datasets exhibit the largest scale variations and the smallest target objects within the RSOC dataset as we show in Appendix B.1. For the RSOC ship and small-vehicle datasets, SI-INR achieves superior counting performance primarily due to its flexibility in generating outputs at arbitrary resolutions. Compared with other methods with fixed downsample ratio and can only generate $64 \times 64$ density maps, SI-INR improves the counting performance by directly generating larger and clearer density maps as we showed in Table 5. These results highlight the significant advantages of SI-INR in handling targets across varying scales.

We visualize the predicted density maps generated by SI-INR and other SOTA methods in Figure 2, excluding eFreeNet (Huang et al., 2023), as it is a regression-based counting method. All four images are randomly sampled from the RSOC dataset, with the first three from the RSOC ship dataset and the last one from the RSOC small-vehicle dataset. As shown, SI-INR delivers more accurate counting performance, particularly when the objects' appearance and scales are complex. In the first three images, SI-INR generates clearer density maps. In the last image from the RSOC small-vehicle dataset, where the cars are too small for the other SOTA methods to detect, SI-INR's scale-invariant property enables it to produce higher-quality density maps and achieve better counting accuracy.

**Inference Efficiency.** In the inference stage of the RSOC building dataset, ASPD-Net requires approximately 15.13 seconds, PSGC-Net takes around 2.47 seconds, eFreeNet takes around 3.84 seconds, and our SI-INR model requires about 3.87 seconds. SI-INR does take longer during the training phase compared to PSGC-Net due to the integration of scale-equivariant models and the use of stacks of linear layers in the INR. However, the inference cost remains acceptable thanks to the simple design of our INR decoder, which consists of only 4 linear layers, and our lightweight scale-equivariant backbone.

Table 1: Comparison of counting performances on the RSOC datasets.

| Method | Loss | | Building | | Small-vehicle | | Large-vehicle | | Ship | |
|--------|------|------|------|------|------|------|------|------|------|------|
| | MSE. | Bayes. | MAE | RMSE | MAE | RMSE | MAE | RMSE | MAE | RMSE |
| MCNN | ✓ | | 13.65 | 16.56 | 488.65 | 1317.44 | 36.56 | 56.55 | 263.91 | 412.30 |
| eFreeNet | ✓ | | 6.99 | 9.61 | 195.86 | 463.62 | 14.55 | 19.77 | 65.34 | 85.45 |
| PSGCNet | ✓ | | 7.33 | 11.02 | 346.78 | 952.64 | 21.54 | 32.75 | 75.27 | 94.79 |
| PSGCNet | | ✓ | 7.18 | 10.98 | 196.25 | 360.15 | 14.47 | 26.19 | 72.07 | 98.06 |
| ASPDNet | ✓ | | 7.40 | 11.06 | 378.23 | 978.93 | 18.76 | 31.06 | 63.32 | 84.85 |
| ASPDNet | | ✓ | 7.59 | 11.07 | 365.69 | 1101.25 | 16.61 | 29.26 | 64.82 | 89.24 |
| SI-INR | | ✓ | 6.54 | 9.80 | 157.18 | 306.43 | 12.61 | 21.78 | 59.76 | 81.79 |

Table 2: Comparison of counting performances on the CARPK and PUCPR+ datasets.

| Method | Loss | | CARPK | | PUCPR+ | |
|--------|------|------|------|------|------|------|
| | MSE. | Bayes. | MAE | RMSE | MAE | RMSE |
| MCNN (Zhang et al., 2016) | ✓ | | 24.95 | 39.63 | 21.86 | 29.53 |
| eFreeNet (Huang et al., 2023) | ✓ | | 46.42 | 52.34 | 18.98 | 23.03 |
| PSGCNet (Gao et al., 2022) | | ✓ | 11.07 | 14.55 | 3.87 | 4.86 |
| PSGCNet (Gao et al., 2022) | | ✓ | 7.71 | 10.28 | 3.17 | 5.27 |
| ASPDNet (Gao et al., 2020) | ✓ | | 10.01 | 12.84 | 4.21 | 5.02 |
| ASPDNet (Gao et al., 2020) | | ✓ | 9.98 | 13.19 | 4.48 | 5.93 |
| SAFECount (You et al., 2023) | ✓ | | 5.33 | 7.04 | 2.24 | 3.44 |
| SI-INR | | ✓ | 5.54 | 7.43 | 2.09 | 2.70 |

Generalization Results. We evaluate the robustness of SI-INR and baseline methods to the scale variation in testing data by testing models using images with resolutions different from training images. To further increase the scale variance inside the RSOC dataset, we keep the training images in a fixed scale $512 \times 512$, rescale test images into 5 different resolutions: $205 \times 205$, $307 \times 307$, $410 \times 410$, $512 \times 512$, $614 \times 614$, and test our SI-INR as well as baselines on the rescaled test images. The varying scales in the test set better reflect real-world conditions, where objects may appear at different sizes due to changes in altitude, camera settings, or image cropping. SI-INR significantly outperforms all baseline models when the variation of resolution in test image is presented. The performance advantages over baseline models illustrate that our SI-INR is not only more robust compared to other baselines when processing images with unseen resolutions, but also more data efficient. Especially on PUCPR+ dataset, SI-INR reduces MAE by 70.9% and RMSE by 71.64% compared with efREENet. Furthermore, PSGCNet employs a traditional pyramidal architecture to address object scale variance in object counting. However, as illustrated in Figure 3 and Figure 5, SI-INR achieves superior counting accuracy and produces higher-quality density maps when facing different resolution inputs even with extremely low resolution like $104 \times 104$, demonstrating its enhanced ability to handle scale variance compared to traditional methods.

As shown in Figure 3 and Figure 5, SI-INR achieves a relative scale-invariance model while truly scale-invariance model is impractical for SESN, SESN relies on group convolutions to approximate scale-equivariance. However, the finite set of sampled scales used during training and inference means that scale-equivariance is not exact but rather approximate within a certain range of scales. Incorporating exact scale-equivariance for all scales would require infinite representations, which is computationally infeasible.

Showcases. We further visualize these results in Figure 3. Although the performances of all the models degrade with small-scale inputs, our SI-INR can still produce density maps with the objects well separated. Moreover, the underestimation of object counts is much less severe in SI-INR compared to all the baselines, which demonstrates the robustness of SI-INR under scale variation. We provide additional results in Appendix B.3.

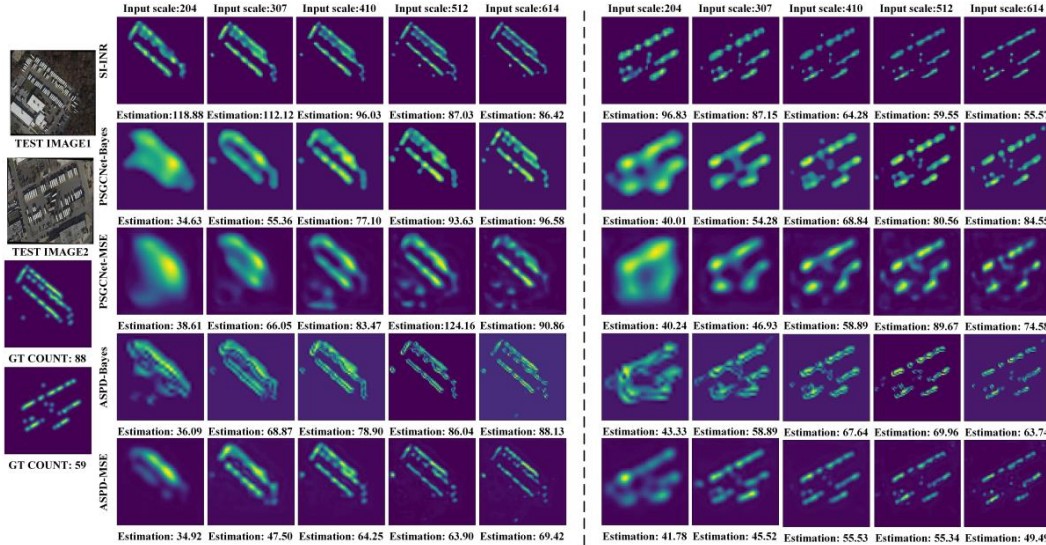

Figure 3: Predicted density maps by SI-INR and other baselines for two test images from RSOC. Two test images are rescaled to $205 \times 205$, $307 \times 307$, $410 \times 410$, $512 \times 512$, $614 \times 614$ before fed into the models.

Table 3: Counting performance of handling unseen scales images on the CARPK, PUCPR+, RSOC building datasets and RSOC large-vehicle datasets.

| Method | Loss | | CARPK | | PUCPR+ | | Building | | Large-vehicle | |
|--------|------|------|------|------|------|------|------|------|------|------|
| | MSE. | Bayes. | MAE | RMSE | MAE | RMSE | MAE | RMSE | MAE | RMSE |
| eFreeNet | ✓ | | 50.37 | 56.83 | 30.45 | 35.59 | 16.97 | 19.72 | 49.39 | 57.26 |
| PSGCNet | ✓ | | 39.36 | 54.13 | 49.51 | 77.05 | 11.56 | 16.43 | 28.74 | 46.22 |
| PSGCNet | | ✓ | 37.63 | 52.46 | 32.58 | 52.38 | 12.09 | 16.96 | 22.47 | 39.99 |
| ASPDNet | ✓ | | 41.28 | 53.62 | 46.31 | 75.18 | 11.31 | 15.60 | 26.86 | 40.17 |
| ASPDNet | | ✓ | 37.25 | 52.26 | 35.02 | 63.90 | 11.37 | 16.11 | 22.11 | 39.37 |
| SI-INR | | ✓ | 24.30 | 28.09 | 8.85 | 11.91 | 7.96 | 11.29 | 21.89 | 30.47 |

## 4.3 Ablation studies

In this section, we evaluate the effect of each constituting component, test the sensitivity of the counting performance of our SI-INR model with respect to Sampling Rate $S_{INR}$. We further discuss the effect of different scale-equivariant models in Appendix B.4.

Effect of constituting components. We conduct ablation experiments to study the effect of each SI-INR component using the RSOC large-vehicle dataset. The results are reported in Table 4. We observe progressive counting performance improvement by introducing each of our model components, which shows that all of the scale-equivariant backbone, hybrid pyramidal scale module, and INR-decoder help improve the counting accuracy.

Table 4: Contributions of each component (SE-backbone, hybrid pyramidal scale module, INR-decoder) in SI-INR.

| Method | RSOC-Large-vehicle | |
|--------|------|------|
| | MAE | RMSE |
| VGG19 | 20.26 | 32.75 |
| VGG19+HPSM+INR | 14.70 | 25.74 |
| SE-Backbone+HPSM | 15.65 | 25.43 |
| SE-Backbone+INR | 16.28 | 28.87 |
| SE-Backbone+HPSM+INR | 12.61 | 21.78 |

Table 5: Effect of sampling rate $S_{INR}$ on object counting in SI-INR.

| Method | RSOC-ship | | RSOC-small-vehicle | |
|---|---|---|---|---|
| | MAE | RMSE | MAE | RMSE |
| $S_{INR}$=8 | 127.14 | 173.21 | 277.50 | 1017.56 |
| $S_{INR}$=16 | 65.49 | 93.03 | 273.78 | 1016.11 |
| $S_{INR}$=32 | 62.97 | 86.81 | 255.12 | 821.21 |
| $S_{INR}$=64 | 62.26 | 83.98 | 243.66 | 731.51 |
| $S_{INR}$=128 | 59.76 | 81.79 | 157.18 | 306.43 |

Effect of sampling rate $S_{INR}$. We also report the prediction accuracy and include the predicted density maps by our SI-INR trained with the loss estimated by sampling from the grids of different size $S_{INR} \times S_{INR}$ in Table 5. In this section, we evaluate the counting performance of SI-INR on the RSOC ship and RSOC small-vehicle datasets with $S_{INR} = 8, 16, 32, 64, 128$. The results show that SI-INR achieves better counting performance as $S_{INR}$ increases from 8 to 128. This effect is particularly pronounced for the RSOC small-vehicle dataset, where the sampling ratio significantly impacts counting accuracy. Since the targets are very small, training with a higher sampling ratio helps the model more accurately locate the vehicles. Besides, this continuous property helps make it easy to balance the computation costs and the counting accuracy requirement. We further visualize several results in Appndix B.7.

## 5 Conclusions

In this paper, we introduce SI-INR, a novel scale-invariant INR implementation that maps discrete grid image signals into continuous 2D function space, maintaining invariance to scaling variation of the input signals. For object counting, SI-INR achieves SOTA performance, our experiments demonstrate that SI-INR is exceptionally robust and flexible compared to existing methods, capable of processing images of unseen resolutions during testing and effectively handling images of various scales during training. This flexibility allows SI-INR to learn and capture more detailed features from different input training images. SI-INR can be easily applied to other image analysis tasks to achieve arbitrary-scale SOTA performance robustly with respect to input image size/resolution. Future work will focus on applying SI-INR to multi-task scenarios, integrating object detection, image segmentation, and depth estimation alongside counting.

## 6 Reproducibility Statement

We have ensured the reproducibility of our experiments by providing detailed descriptions of the model architectures, data augmentation steps, training procedures and hyperparameters setup in Section 4.1. The datasets are introduced in Appendix B.1. For baseline models, we also give training details and hyperparameter configurations in Appendix B.2. Additionally, the code is included in the Supplementary Material and will be made publicly available in a repository to support further verification and replication by other researchers.

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

## A    Supplementary information of model construction

### A.1    Scale equivariance and invariance

Consider a Scale-Translation Group consisting of a Scaling Group $G_S$ and a Translation Group $G_T$, $H = \{h = (s,t) \,|\, s \in G_S, t \in G_T\}$, where $h$ denotes an element of $H$ and represents one scale-translation operator, $G_S$ denotes the Scaling Group, which accounts for transformations that scale an object or function, and $G_T$ denotes the Translation Group, which handles shifting the object or function within its domain. Besides, $s$ is the scaling parameter, indicating how the input is stretched or compressed; $t$ is the translation parameter, specifying the shifting in the domain.

From the group theory, given an image $I_a \in V_1$, a mapping $\Phi : V_1 \to V_2$ is scale-equivariant if:

$$\Phi(p_1(h)(I_a)) = p_2(h)(\Phi(I_a)), \tag{12}$$

where $V_2$ denotes the output domain, $p_1(\cdot)$ and $p_2(\cdot)$ denote the corresponding group actions of $h$ acting on $V_1$, $V_2$. If $p_2(h)$ is the identity mapping, the mapping $\Phi$ is scale-invariant.

### A.2    Scale-translation equivariance of SESN

Consider a steerable convolution filter (Sosnovik et al., 2019), $\varphi_m(x) = m^{-1}\varphi(m^{-1}x)$, which has the following property:

$$p(s^{-1})(\varphi_m)(x) = \varphi_m(sx) = s^{-1}\varphi_{s^{-1}m}(x), \tag{13}$$

where $p(s)$ denotes the group action of $s$ on convolution filters. The scaling of this filter is the transformation of its parameters.

With scale-translation group $H$ and steerable convolution filters $\varphi_m(\cdot)$, the group-equivariant convolution on $f$ can be defined as:

$$
\begin{aligned}
[f \star_H \psi_m](s, t) &= \int_S \int_T f(s', t') \, p(s, t) \, [\psi_m](s', t') \, d\mu(s') \, d\mu(t') \\
&= \sum_{s'} \int_T f(s', t') \, \psi_{sm}(s^{-1}s', t' - t) \, dt' = \sum_{s'} \left[ f(s', \cdot) \star \psi_{sm}(s^{-1}s', \cdot) \right](t).
\end{aligned}
\tag{14}
$$

The proof of translation-equivariance of Equation (14) is as follows:

$$
\begin{aligned}
[p(\hat{t})[f] \star_H \psi_m](s, t) &= \sum_{s'} \left[ p(\hat{t})[f](s', \cdot) \star \psi_{sm}(s^{-1}s', \cdot) \right](t) \\
&= \sum_{s'} p(\hat{t}) \left[ f(s', \cdot) \star \psi_{sm}(s^{-1}s', \cdot) \right](t) \\
&= p(\hat{t}) \left\{ \sum_{s'} \left[ f(s', \cdot) \star \psi_{sm}(s^{-1}s', \cdot) \right] \right\}(t) \\
&= p(\hat{t}) \left[ f \star_H \psi_m \right](s, t).
\end{aligned}
\tag{15}
$$

The proof of scale-equivariance of Equation (14) can be obtained:

$$
\begin{aligned}
[p(\hat{s})[f] \star_H \psi_m](s, t) &= \sum_{s'} \left[ p(\hat{s})[f](s', \cdot) \star \psi_{sm}(s^{-1}s', \cdot) \right](t) \\
&= \sum_{s'} p(\hat{s}) \left[ f(\hat{s}^{-1}s', \cdot) \star \psi_{\hat{s}^{-1}sm}(s^{-1}s', \cdot) \right](t) \\
&= \sum_{s''} \left[ f(s'', \cdot) \star \psi_{\hat{s}^{-1}sm}(\hat{s}s^{-1}s'', \cdot) \right](\hat{s}^{-1}t) \\
&= [f \star_H \psi_m](\hat{s}^{-1}s, \hat{s}^{-1}t) \\
&= p(\hat{s}) \left[ f \star_H \psi_m \right](s, t).
\end{aligned}
\tag{16}
$$

Finally, we have the proof of scale-translation equivariance of Equation (14):

$$
\begin{aligned}
p(\hat{s}\hat{t})[f] \star_H \psi_m &= p(\hat{s})p(\hat{t})[f] \star_H \psi_m = p(\hat{s}) \left[ p(\hat{t})[f] \star_H \psi_m \right] \\
&= p(\hat{s})p(\hat{t}) \left[ f \star_H \psi_m \right] = p(\hat{s}\hat{t}) \left[ f \star_H \psi_m \right].
\end{aligned}
\tag{17}
$$

The summation of two scale-equivariant models is still scale-equivariant by the following Lemma.

Lemma 1 The summation of two equivariant mappings $\Phi_1 : V_1 \to V_2$, $\Phi_2 : V_1 \to V_2$ is still equivariant.

Proof: For any scale-translation operator $h$ we have:

$$
\begin{aligned}
(\Phi_1 + \Phi_2)(p_1(h)(\mathrm{I}_a)) &= \Phi_1(p_1(h)(\mathrm{I}_a)) + \Phi_2(p_1(h)(\mathrm{I}_a)) \\
&= p_2(h)(\Phi_1(\mathrm{I}_a)) + p_2(h)(\Phi_2(\mathrm{I}_a)) = p_2(h)((\Phi_1 + \Phi_2)(\mathrm{I}_a)).
\end{aligned}
\tag{18}
$$

### A.3 Derivation of the minimization objective $\mathcal{L}_{ELBO}$

Here we detail the process of deriving the minimization objective in Section 3.3 of the Main Text.

For a given image $\mathbf{I}$, we define the counting annotation map as $\mathcal{D}_{\mathbf{I}} = \{(\mathbf{m}_n, y_n)\}_1^N$. Where $\mathbf{m}_n \in [0, 1]^2$ denotes the normalized image-coordinate position of the $n$-th object, $y_n = n$ is

the corresponding label for each object, and $N$ denotes the total number of labeled objects in $\mathbf{I}$. Typically, people generate ground truth density maps $\mathbf{D}^{gt}$ by convolving this annotation map with a Gaussian kernel (Fu et al., 2015; Paul Cohen et al., 2017; Gao et al., 2020).

Following Ma et al. (2019), we define $y(\cdot) : \mathbb{R}^2 \to \{1, \cdots, N\}$, with $y(\mathbf{x})$ as whether the location $\mathbf{x}$ belongs to one of the $N$ objects computed by a prior distribution $p(y(\cdot))$. The posterior distribution can be expressed as:

$$p\left(y(\cdot) = n \mid \mathcal{D}_{\mathbf{I}_a}\right) = \frac{\mathcal{N}\left(\cdot; \mathbf{m}_n, \sigma^2 \mathbf{1}_{2 \times 2}\right)}{\sum_{i=1}^{N} \mathcal{N}\left(\cdot; \mathbf{m}_i, \sigma^2 \mathbf{1}_{2 \times 2}\right)}. \tag{19}$$

Based on the definition of $c_n^{gt}(\cdot)$ in Ma et al. (2019), the contribution in the likelihood can be expressed as follows:

$$c_n^{gt}(\cdot) = p\left(y(\cdot) = n \mid \mathcal{D}_{\mathbf{I}_a}\right) \times \mathbf{D}^{\mathrm{gt}}(\cdot) = \frac{\mathcal{N}\left(\cdot; \mathbf{m}_n, \sigma^2 \mathbf{1}_{2 \times 2}\right)}{\sum_{i=1}^{N} \mathcal{N}\left(\cdot; \mathbf{m}_i, \sigma^2 \mathbf{1}_{2 \times 2}\right)} \times \sum_{i=1}^{N} \mathcal{N}\left(\cdot; \mathbf{m}_i, \sigma^2 \mathbf{1}_{2 \times 2}\right)$$
$$= \mathcal{N}\left(\cdot; \mathbf{m}_n, \sigma^2 \mathbf{1}_{2 \times 2}\right), \tag{20}$$

We can find that $c_n^{gt}(\cdot)$ is a Gaussian distribution centered at $\mathrm{m}_n$, and the summation of $c_n^{gt}(\cdot)$ equals one.

Thanks to the continuous property of SI-INR, Bayesian counting loss can be represented as:

$$\mathcal{L}_{BAY} = \frac{1}{A} \sum_{a=1}^{A} \Big[ \sum_{n=0}^{N} (\mathbb{E}_{\mathbf{x} \sim p_s(\mathbf{x})}(c_{n,a}) - \mathbb{E}_{\mathbf{x} \sim p_s(\mathbf{x})}(c_{n,a}^{\mathrm{gt}})) \Big]. \tag{21}$$

where $p_s(\mathbf{x})$ is any probability distribution of $\mathbf{x}$. This optimization objective enables our model to be trained using any existing stochastic optimization algorithm. Without loss of generality, we can select $p(\mathbf{x})$ as a uniform distribution, $\mathbf{x} \sim \mathrm{Uniform}[0, 1]^2$.

Besides, we add MAE counting loss into our loss function, the final minimization objective is:

$$\mathcal{L}_{SI-INR} = \frac{1}{A} \sum_{a=1}^{A} \Big[ \sum_{n=0}^{N} (\mathbb{E}_{\mathbf{x} \sim \mathrm{Unif}[0,1]^2}(c_{n,a}) - \mathbb{E}_{\mathbf{x} \sim \mathrm{Unif}[0,1]^2}(c_{n,a}^{\mathrm{gt}}))$$
$$+ \kappa (\sum_{n=0}^{N} (\mathbb{E}_{\mathbf{x} \sim \mathrm{Unif}[0,1]^2}(c_{n,a})) - N) \Big] \tag{22}$$

## B Supplementary information of experiments

### B.1 Data

RSOC    (Gao et al., 2020): The Remote Sensing Object Counting (RSOC) dataset is a large-scale benchmark specifically designed for counting objects in satellite imagery. It includes a total of $3,057$ images with $286,539$ annotated object instances. The dataset is divided into four distinct subdatasets, each focused on a different object type: Buildings, Small Vehicles, Large Vehicles, and Ships. The RSOC Buildings dataset contains 1,205 training images and 1,263 test images, where the image resolution is $512 \times 512$. The RSOC Small Vehicles dataset has 222 training images and 58 test images. The image resolution ranges from $421 \times 799$ to $12029 \times 5014$. Large Vehicles consists of 108 training images and 64 test images, The image resolution ranges from $731 \times 596$ to $6327 \times 5662$. And the Ships subset has 97 training images and 60 test images. The image resolution ranges from $606 \times 1065$ to $6335 \times 3591$.

CARPK    (Hsieh et al., 2017): The Car Parking Lot Dataset (CARPK) is a benchmark for car counting tasks, consisting of 1,148 images taken from drone perspectives over four parking lots, containing 89,777 annotated cars. These images capture real-world scenarios

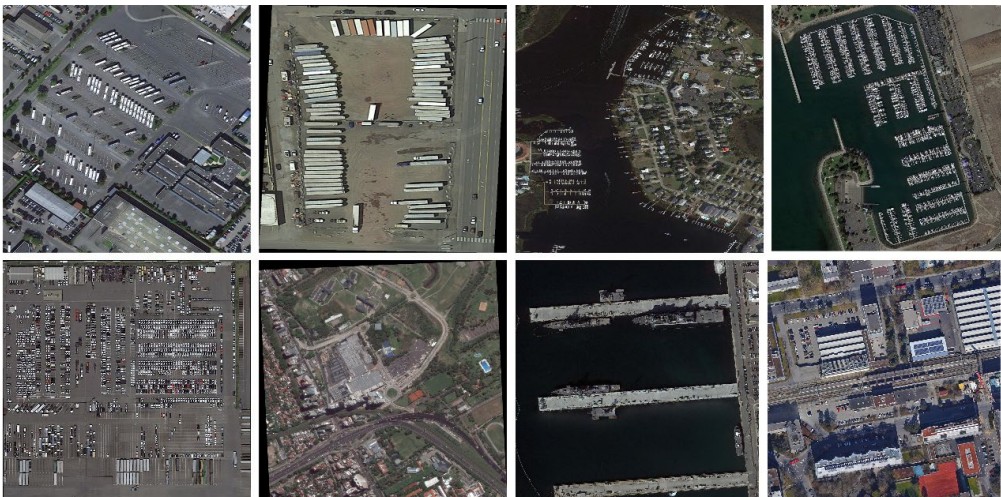

Figure 4: Example images from the RSOC dataset.

with dense vehicle arrangements, making the dataset challenging for object detection and counting tasks. The average resolution of the images is $1280 \times 720$ pixels, providing detailed aerial views. Each image is annotated with bounding boxes around individual cars, making the dataset suitable for both object counting and detection. The dataset is split into 989 training images and 459 testing images.

PUCPR+ (Hsieh et al., 2017): The Pontifical Catholic University of Parana+ Dataset (PUCPR+) is a specialized car counting resource where all images are captured from the 10th floor of a building. PUCOR+ contains 125 images with $16,456$ cars, where 100 images are set for training, while the remaining images are utilized for testing the models.

Visualization We further provide several exemplar images from RSOC datasets in Figure 4. It can be found that the objects within the same image naturally appear of similar size. However, remote sensing datasets, including RSOC, encompass images with a wide range of resolutions. As a result, object sizes vary significantly across different images, even if they appear uniform within a single image. Furthermore, we resize images to various resolutions to evaluate robustness to scale variability which further increases the range of scale differences across different images in our experiments. In Figure 4, the top-left two images are both from the RSOC large-vehicle dataset, clearly showing that the cars in the second image are three times larger than those in the first image. Similarly, the bottom-left two images, from the RSOC small-vehicle dataset, highlight the differences in visibility: cars are clearly seen in the first image but are almost invisible in the second.

### B.2 Baselines

In this section, we delve into the training specifics for all baseline models utilized in our experiments.

ASPDNet (Gao et al., 2020): ASPDNet is an advanced attention-based network that integrates scale pyramids and deformable convolutions to effectively utilize attention mechanisms. This architecture captures extensive contextual and high-level semantic information, which aids in reducing the impact of cluttered backgrounds while emphasizing the regions of interest. In our study, we follow its original network design, set the batch size as 16, replace the original Stochastic Gradient Descent (SGD) optimizer with ADAM (Kingma & Ba, 2014) optimizer, and set the learning rate as $1e-4$ to enhance the training results. Our training spans 200 epochs. ASPDNet is trained under MSE counting and Bayes counting (Ma et al., 2019) respectively to get the best counting performance.

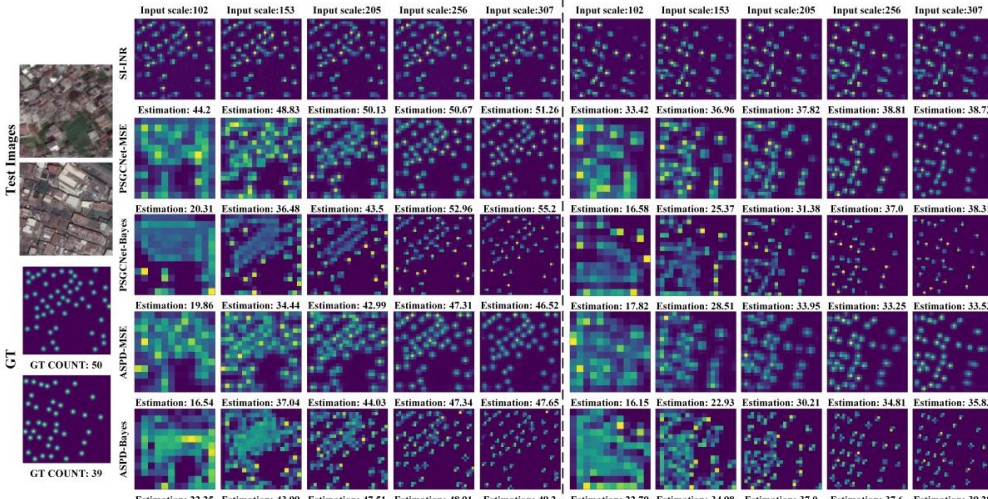

Figure 5: Predicted density maps by SI-INR and other baselines for two test images from RSOC. Two test images are rescaled to $102 \times 102$, $153 \times 153$, $205 \times 205$, $256 \times 256$, $307 \times 307$ before fed into the models.

PSGCNet (Gao et al., 2022): PSGCNet integrates pyramidal scale and global context modules to handle scale variations of remote sensing images. We follow the network setup, setting the learning rate as $1e - 4$, and using a batch size of 16. We trained PSGCNet with original Bayesian-based counting loss and MSE respectively. Our training spans 200 epochs.

eFreeNet (Huang et al., 2023): The eFreeNet is an ensemble of first-rank-then-estimate networks that tailors a ranking metric optimization scheme to fit object counting. The study employs the default network architecture. In the optimization setup, we set the backbone's learning rate as $1e - 5$ while $1e - 5$ for other components following the original setup. The ensemble number is set as 8 to get the best counting performance. Our training spans 3000 epochs with a batch size of 8.

### B.3 Additional qualitative results

We provide additional qualitative results here. For the figures from the RSOC building dataset, two test images are rescaled to $102 \times 102$, $153 \times 153$, $205 \times 205$, $256 \times 256$, and $307 \times 307$ before being input into the models. The results in Figure 5 highlight not only the counting accuracy of our model but also its robustness and ability to handle inputs of varying resolutions.

### B.4 Effect of different Scale-equivariant models

In our experiments, we test SESC (Sosnovik et al., 2019) and scale-equivariant Fourier layers (Rahman & Yeh, 2023) in SI-INR, Compared with SESC, scale-equivariant Fourier layers demand more computational resources, especially processing images over 512 resolutions. On the RSOC building dataset, training one epoch for SI-INR with SESC takes 161 seconds, compared with ASPDNet's 167 seconds and PSGCNet's 121 seconds. However, SI-INR with scale-equivariant Fourier layers takes more than 900 seconds, which is close to one order of magnitude more costly.

### B.5 Comparison on UCF-QNRF dataset

We compare our SI-INR with state-of-the-art methods as well as our baselines on the UCF-QNRF (University of Central Florida - Qatar National Research Fund) dataset (Idrees et al., 2018), which is a highly diverse dataset consisting of 1,535 images with over 1.2 million annotated individuals, spanning a wide range of crowd densities and changing object sizes. We report the results in Table 6.

Table 6: Performance Comparison on the UCF-QNRF Dataset

| Model | MAE | RMSE |
|---|---|---|
| MMNet (Dong et al., 2020) | 104.00 | 178.00 |
| MSFFA (Li et al., 2023) | 94.60 | 170.60 |
| MFANet (Zhu et al., 2021) | 97.7 | 166.00 |
| CLTR (Liang et al., 2022) | 85.80 | 141.30 |
| Bayesian+ (Ma et al., 2019) | 88.70 | 154.80 |
| P2PNet (Song et al., 2021) | 85.32 | 154.50 |
| GauNet (Cheng et al., 2022) | 81.60 | 153.71 |
| APGCC (Chen et al., 2025) | 80.10 | 136 .60 |
| PSL-Net (Ryu & Song, 2024) | 85.50 | 144.40 |
| PET (Liu et al., 2023) | 79.53 | 144.32 |
| PSGCNet (Baseline) | 86.30 | 149.50 |
| SI-INR (Ours) | 80.89 | 134.73 |

The reported results indicate that SI-INR achieves competitive performance, with a Mean Absolute Error (MAE) of 80.89 and a Root Mean Squared Error (RMSE) of 134.73. Additionally, Compared with existing density map based methods, such as Bayesian+ (Ma et al., 2019), GauNet (Cheng et al., 2022), and our baseline PSGCNet (Gao et al., 2022), SI-INR demonstrates consistent improvements in both metrics. These results highlight the effectiveness of our proposed approach.

### B.6 Effect of different methods for handling multi-scale challenges

For efficiency and adaptability, scale-equivariant methods adjust to different scales without having separate filters for each scale, unlike traditional methods that may rely on resizing inputs or using multiple filters for different scales. Many traditional multi-scale methods, such as image pyramids or multi-resolution networks, may struggle with high computational costs because they process the same image at multiple resolutions, leading to increased complexity especially with large images or when handling many scales. Besides, traditional multi-scale approaches do not focus on deriving scale-invariant outputs compared with our SI-INR.

In our experiment, we have compared our SI-INR with four SOTA methods (PSGCNet (Gao et al., 2022), MMNet (Dong et al., 2020), MFANet (Zhu et al., 2021), MSFFA (Li et al., 2023)) that mainly focus on handling multi-scale challenges. PSGCNet applies a pyramidal network to handle multi-scale challenges, MMNet leverages multi-level density-based spatial information, MFANet introduces multi-level feature aggregation, and MSFFA integrates multi-scale feature fusion and attention mechanisms. As we report in Table 6, our SI-INR outperforms these methods on the UCF-QNRF crowd-counting dataset, which demonstrates its superior performance in handling objects of different sizes.

### B.7 Visualization of different sampling rate $S_{INR}$ of SI-INR

To further demonstrate the effect of sample rate $S_{INR}$ of SI-INR, we visualize SI-INR's outputs when setting the sample rate from 8 to 128 in the Figure 6. In this ablation experiment, we let the well-trained SI-INR model directly generate 5 different resolution density maps, we can find that SI-INR can generate high-quality density maps when the sample rate sampling rate $S_{INR}$ increases.

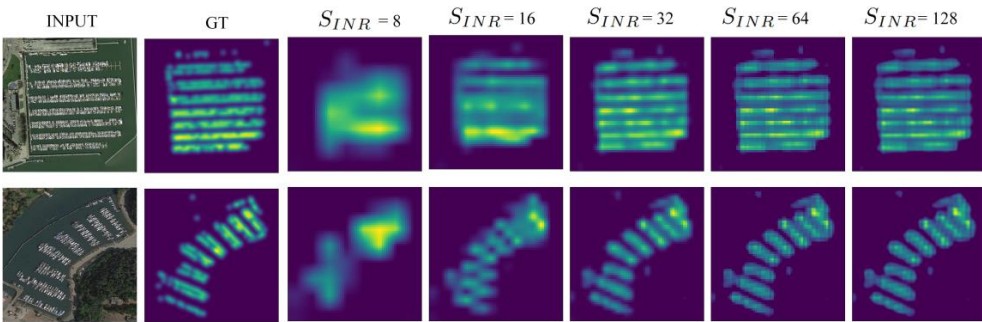

Figure 6: Predicted density maps by SI-INR with different sampling rate $S_{INR}$.

