# OpenReview forum: "Scale-Invariant Continuous Implicit Neural Representations For Object Counting"
_ICLR.cc/2025/Conference — Submitted to ICLR 2025_

### Official Review · Reviewer_fHTu · 2024-10-29

**Soundness:** 3
**Presentation:** 3
**Contribution:** 3
**Rating:** 6
**Confidence:** 5

**Summary:**

This paper introduces a framework, Scale-Invariant Implicit Neural Representations (SI-INR), for object counting across varying image scales and resolutions. SI-INR addresses limitations in current methods, particularly the challenge of scale invariance in object counting tasks involving dense object scenes. The approach combines a scale-equivariant backbone with implicit neural representations, achieving high accuracy across benchmarks like the RSOC and CARPK datasets.

**Strengths:**

- The framework’s focus on handling scale invariance through a continuous mapping function and modular structure is particularly valuable for applications involving heterogeneous datasets or remote sensing imagery with objects of varying sizes.

- Experiments across multiple datasets, such as RSOC and CARPK, showcase the generalizability of SI-INR. The paper also provides sufficient details on network configurations, making the work reproducible.

**Weaknesses:**

- The paper does not fully explain the operational details of the "scale-invariant continuous mapping," especially regarding how SI-INR preserves fine details for low-resolution inputs. A more comprehensive description would enhance reproducibility.

- While computational demands are briefly mentioned, there is no clear comparison of SI-INR's runtime performance against baselines in various resolutions. This omission makes it difficult to assess scalability for real-time applications.

- The paper could benefit from additional experiments testing SI-INR’s robustness with images at extreme resolutions, especially low resolutions (e.g., <200 pixels), to provide a more complete understanding of its limitations.

- Missing some important previous work:

[1] Learning spatial awareness to improve crowd counting. (2019). ICCV 2019

[2] Rethinking spatial invariance of convolutional networks for object counting. (2022).CVPR 2022

**Questions:**

- Could you provide further clarification on the continuous mapping mechanism in SI-INR? Specifically, how does it handle low-resolution inputs without sacrificing accuracy?

- Would you consider evaluating SI-INR’s performance in sequential data or video-based counting tasks?

- Have you identified any potential biases in SI-INR when applied to diverse environmental conditions, such as different lighting or weather effects in remote sensing?

---

> ### Author Response · Authors · 2024-11-19
> **Rebuttal for Reviewer fHTu(1/2)**
>
> **1) The paper does not fully explain the operational details of the "scale-invariant continuous mapping," especially regarding how SI-INR preserves fine details for low-resolution inputs. A more comprehensive description would enhance reproducibility:**
>
> Thank you for your valuable feedback. We appreciate your observation regarding the need for more details about the "scale-invariant continuous mapping" and its role in preserving fine details for low-resolution inputs.
>
> The "scale-invariant continuous mapping" in SI-INR relies on the scale-equivariant/invariant properties of each network component and the implicit neural representation (INR) model to learn a continuous function that maps coordinates to density values.
>
> Specifically, a scale-equivariant Backbone B which satisfies $B(p_1(h) (\textbf{I}_a)) = p_B(h)(B)(\textbf{I}_a)$ is adopted to extract deterministic features so that scale changes in objects will only affect the scale of feature maps while keeping the appearance. Later, a scale-invariant encoder $E$ maps the features into a constant latent space and an INR decoder is applied to transform the latent into a continuous function. $\Psi(p_1(h) (\textbf{I}_a))(\mathbf{x}) = \mathcal{H}(E(B(p_1(h) (\textbf{I}_a)))(\mathbf{x}) = \mathcal{H}(E(B(\textbf{I}_a)))(\mathbf{x}) = \Psi(\textbf{I}_a)(\mathbf{x})$, where $\Psi(\cdot)$ denotes in total SI-INR model, $\mathcal{H}(\cdot)$ represent our INR decoder.
>
> For low-resolution inputs, the scale-equivariant property of our backbone B enables SI-INR to generate outputs that are much closer to the results obtained from the same input at a larger scale, outperforming traditional methods.
>
> Traditional approaches, which often use fixed downsampling ratios, produce low-resolution density maps for low-resolution images. The corresponding unclear, low-resolution discrete density ground truth is less effective for training. In contrast, our SI-INR is trained using high-resolution ground-truth density maps, even for low-resolution inputs, avoiding significant information loss. This design enables SI-INR to capture spatial relationships beyond the input's native resolution and preserve fine details effectively.
>
> To enhance reproducibility, we will include a more detailed explanation of the "scale-invariant continuous mapping" in the revised manuscript. This will cover the training process, the role of the INR model, and the mechanisms that help retain fine details for low-resolution inputs. Additionally, we will provide visual illustrations to clarify these concepts.
>
> **2) While computational demands are briefly mentioned, there is no clear comparison of SI-INR's runtime performance against baselines in various resolutions. This omission makes it difficult to assess scalability for real-time applications:**
>
> We appreciate the reviewer’s suggestion of including the runtime comparison. To address this, we have conducted a speed test using a workstation with an NVIDIA V100 32GB GPU. For the RSOC small-vehicle dataset, we observed the following inference times: ASPD-Net requires approximately 15.13 seconds, PSGC-Net takes around 2.47 seconds, eFreeNet takes around 3.84 seconds, and our SI-INR model requires about 3.87 seconds.
>
> | Model        | Inference Time (seconds) |
> |--------------|---------------------------|
> | eFreeNet     | 3.84                      |
> | ASPD-Net     | 15.13                     |
> | PSGC-Net     | 2.47                      |
> | SI-INR (Ours)| 3.87                      |
>
> SI-INR requires more time for inference compared to PSGC-Net, due to the integration of the scale-equivariant models and the stacks of linear layers in the INR. However, thanks to our lightweight scale-equivariant backbone and the compact design of the INR model, which consists of only 4 linear layers, the inference cost remains manageable and acceptable. We will include these findings in the revised manuscript for clearer scalability discussion. We further give a more detailed discussion in our Rebuttal for all reviewers.

---

> ### Author Response · Authors · 2024-11-20
> **Rebuttal for Reviewer fHTu(2/2)**
>
> **3) The paper could benefit from additional experiments testing SI-INR’s robustness with images at extreme resolutions, especially low resolutions (e.g., <200 pixels), to provide a more complete understanding of its limitations:**
>
> Thank you for your constructive suggestions. We here add an experiment to evaluate SI-INR’s robustness with low resolutions. On the RSOC building dataset, We resize all test images into $100\times100$ and compare the counting performance of SI-INR and baseline models. PSGCNet achieves an MAE of 18.15 and an MSE of 22.78, APDNet achieves an MAE of 20.04 and an MSE of 24.32, EfreeNet achieves an MAE of 24.41 and an MSE of 27.06. In comparison, our SI-INR significantly outperforms these methods, achieving an MAE of 8.53 and an RMSE of 12.65.
>
> We plan to extend this analysis by testing at additional resolutions and plotting a performance curve to further illustrate the robustness of SI-INR against resolution changes.
>
> **4 Missing some important previous work:**
>
> We truly appreciate the suggestions on comparing SI-INR to important previous works. During the rebuttal period, we have further evaluated SI-INR on the large-scale and challenging UCF-QNRF dataset.
>
> UCF-QNRF comprises 1,535 images with over 1.2 million annotated individuals, capturing a wide range of crowd densities. SI-INR has achieved competitive performance, with an MAE of 80.89 and RMSE of 134.73. For comparison, while P2PNet [1](A point-to-point matching method for crowd counting) achieves an MAE of 85.32 and an RMSE of 154.5, GauNet [2](leveraging Gaussian-based density maps) achieves an MAE of 81.60 and an RMSE of 153.71, APGCC [3](A point-to-point matching method for crowd counting) achieves an MAE of 80.10 and an RMSE of 136.60. PSL-Net [4] achieves an MAE of 85.50 and an RMSE of 144.40. Furthermore, compared to our baseline model, PSGCNet (MAE 86.3, RMSE 149.5).
>
> | Model             | MAE   | RMSE   |
> |-------------------|-------|--------|
> | P2PNet [1]        | 85.32 | 154.50 |
> | GauNet [2]        | 81.60 | 153.71 |
> | APGCC [3]         | 80.10 | 136.60 |
> | PSL-Net [4]       | 85.50 | 144.40 |
> | PSGCNet(baseline) | 86.30 | 149.50 |
> | SI-INR (Ours)     | 80.89 | 134.73 |
>
> The initial results illustrate that our model can indeed achieve comparable performance to the State-of-the-art crowd counting methods on UCF-QNRF dataset. In the final version, we will provide a more comprehensive comparison on additional crowd counting datasets with more baseline methods. For the RSOC, CARPK and PUCPR+ datasets, we will add more result by additional SOTA methods in Table 1 of the main text.
>
> [1] "Rethinking counting and localization in crowds: A purely point-based framework." Proceedings of the IEEE/CVF International Conference on Computer Vision. 2021.
>
> [2] "Rethinking spatial invariance of convolutional networks for object counting." Proceedings of the IEEE/CVF Conference on Computer Vision and Pattern Recognition. 2022.
>
> [3] "Improving Point-based Crowd Counting and Localization Based on Auxiliary Point Guidance." European Conference on Computer Vision. Springer, Cham, 2025.
>
> [4] "Crowd Counting and Individual Localization using Pseudo Square Label." IEEE Access (2024).
>
>
>
> **5) Would you consider evaluating SI-INR’s performance in sequential data or video-based counting tasks?**
>
>
> We appreciate the reviewer's suggestion. We believe that leveraging the scale-equivariant and continuous properties of SI-INR can improve the performance in general, including tracking and counting people across video frames, while also retaining fine spatial details. If given more time, we will evaluate SI-INR's performance in more diverse computer vision tasks in our future research.
>
> **6) Have you identified any potential biases in SI-INR when applied to diverse environmental conditions, such as different lighting or weather effects in remote sensing?**
>
> We focus on addressing challenges due to scale/resolution variations by introduce scale-invariance implicit neural representations. To further overcome the issues under different adversarial conditions as pointed out by the reviewer, different model formulations and solution strategies such as incorporating potential physics models for corresponding lighting or weather effects, which can be potential research for integrating additional such modules. Without explicitly modeling these adversarial effects, we expect that our SI-INR will achieve similarly reported results in the object detection literature. We are open to evaluate these if given more time.

---

### Official Review · Reviewer_dyoE · 2024-10-31

**Soundness:** 3
**Presentation:** 3
**Contribution:** 3
**Rating:** 6
**Confidence:** 4

**Summary:**

This paper introduces a novel framework, Scale-Invariant Implicit Neural Representation (SI-INR), aimed at addressing significant challenges in object counting under varying scales and image resolutions. Traditional CNN-based counting methods often suffer from performance degradation when encountering objects at unseen scales or perspectives due to their reliance on discrete grid representations and non-scale-invariant models. The proposed SI-INR framework leverages a continuous function space, transforming discrete image signals into scale-invariant, continuous representations to improve accuracy and generalizability in object counting tasks.

**Strengths:**

1. The formulas are sufficient, demonstrating a solid mathematical foundation.
2. The method is reasonable.
3. The experiments are comprehensive, proving the effectiveness of the method through empirical validation.

**Weaknesses:**

1. After reading the paper, my understanding is that a VAE-like approach is used to obtain a fixed-size output, ensuring that inputs of different scales can be mapped to the same output, thus addressing the issues mentioned in the paper. However, the paper is written in a very complex manner and requires careful reading to understand. I suggest redrawing Figure 1. First, the new Figure 1 should allow readers to immediately grasp your network, especially the input and output. Additionally, it should include both the overall framework and detailed components of the network. The current Figure 1 does not provide a clear understanding of your work, making it difficult to reproduce.

2. Although your method is reasonable, it does not achieve the result mentioned in line 139 of the paper, i.e., obtaining the same output from inputs of different scales. In previous methods, multi-scale image augmentation was commonly used during training to address this problem. Although your approach is different, the goal remains the same.

3. In line 150, it is mentioned that GT is continuous. Why is GT continuous, whereas the output of previous methods is discrete? And why is your method’s output continuous?

**Questions:**

see weakness

---

> ### Author Response · Authors · 2024-11-19
> **Rebuttal for Reviewer dyoE(1/2)**
>
> **1) After reading the paper, my understanding is that a VAE-like approach is used to obtain a fixed-size output, ensuring that inputs of different scales can be mapped to the same output, thus addressing the issues mentioned in the paper. However, the paper is written in a very complex manner and requires careful reading to understand. I suggest redrawing Figure 1. First, the new Figure 1 should allow readers to immediately grasp your network, especially the input and output. Additionally, it should include both the overall framework and detailed components of the network. The current Figure 1 does not provide a clear understanding of your work, making it difficult to reproduce:**
>
> Thank you for your insightful comments. We appreciate your suggestion to improve Figure 1 and to provide a clearer explanation of the model architecture design.
> To address your concerns, we will redesign Figure 1 to provide a more intuitive illustration of corresponding components in our SI-INR. The revised figure will:
>
> a) clearly depict the input and output of the SI-INR model to help readers understand the scale-normalizing process;
>
> b) present the overall framework alongside detailed components, and how it handles scale variability;
>
> c) include annotations and visual aids to clarify the flow of data through the model.
>
>
> **2) Although your method is reasonable, it does not achieve the result mentioned in line 139 of the paper, i.e., obtaining the same output from inputs of different scales. In previous methods, multi-scale image augmentation was commonly used during training to address this problem. Although your approach is different, the goal remains the same:**
>
> We appreciate the reviewer's comments.  In detail, we choose SESN to achieve the scale-equivariant backbone, SESN relies on group convolutions to approximate scale-equivariance. However, the finite set of sampled scales used during training and inference means that scale-equivariance is not exact but rather approximate within a certain range of scales. Incorporating exact scale-equivariance for all scales would require infinite representations, which is computationally infeasible. Our SI-INR balances computational efficiency with such an exact equivariance guarantee, leading to trade-offs in its ability to generalize across scales. For example, a scale-equivariant steerable convolution will generate an output in shape [$B,S,C_{out},W,H$] for an input in shape [$B,C_{in},W,H$] compared with [$B,C_{out},W,H$] of the traditional convolution. The extra axis $S$ denotes the number of scales sampled from the scaling group. It is clear that a larger $S$ results in a closer approximation to exact scale-equivariance. In SI-INR, we choose 7 different scales from 0.8 to 1.2 to balance the computational efficiency with the exact equivariance, we will add this information in our final version.
>
> Our framework integrates the scale invariance property of object counting into the inherent inductive bias of the model with the SESN encoder, which will guarantee the output being invariant to any change of scale while previous work relies on the heuristic data augmentation method, which may not cover enough range of size variations and does not have any guarantee on size-invariance.
>
> We additionally perform a comparative study between (1) PSGCNet with multi-scale image augmentation and (2) PSGCNet with SI-INR's backbone and INR decoder. On the RSOC building dataset, PSGCNet with multi-scale image augmentation achieves an MAE of 7.04 and an MSE of 10.65, while PSGCNet with our components achieves an MAE of 6.54 and an MSE of 9.80. On the CARPK dataset, PSGCNet with multi-scale image augmentation achieves an MAE of 6.53 and an MSE of 9.41, while PSGCNet with our components achieves an MAE of 5.54 and an MSE of 7.43. We will test on more datasets and methods and add this ablation study to our revised manuscript.

---

> > ### Author Response · Authors · 2024-11-19
> > **Rebuttal for Reviewer dyoE(2/2)**
> >
> > **3) In line 150, it is mentioned that GT is continuous. Why is GT continuous, whereas the output of previous methods is discrete? And why is your method’s output continuous?:**
> >
> > In our approach, the GT is continuous because the traditional density map is defined as a mixture of Gaussian distributions [1]. However, previous methods can only generate discrete outputs thus regress to density maps generated by discretizing the continous mixture of Gaussian density. This process inherently leads to a loss of information.
> >
> > In contrast, our SI-INR leverages the ability of the INR model to learn a continuous function. We provide a more detailed introduction to the continuous representations in our overall rebuttal. The INR model allows us to train the model with continuous density maps using a random sampling algorithm, which preserves more accurate ground-truth information compared to traditional methods. By doing so, our approach introduces a finer level of detail in the density maps, improving the counting performance.
> >
> > [1] "Bayesian loss for crowd count estimation with point supervision." Proceedings of the IEEE/CVF international conference on computer vision. 2019.

---

### Official Review · Reviewer_mauM · 2024-11-03

**Soundness:** 3
**Presentation:** 2
**Contribution:** 2
**Rating:** 3
**Confidence:** 5

**Summary:**

This paper proposes a scale-invariant method that maps discrete grid image signals into a continuous 2D function space. This approach allows the model to represent an image as a continuous function rather than fixed discrete pixels. The model learns through supervised learning to capture and represent scale-invariant features.

**Strengths:**

1. The proposed mehtod achieves better performance than several baseline on some simple datasets.

**Weaknesses:**

1. The comparison methods are very limited; many state-of-the-art methods are not included, such as those using optimal transport and point-to-point matching.
2. As shown in Figure 2, the objects are of similar size, making it difficult to justify that the method effectively addresses the multi-scale challenge.
3. The proposed method should be evaluated on a typical dataset with dense crowds, which are known to have multi-scale individuals.
4. The description of the scale-invariant model is unclear.

**Questions:**

1. Why can it be used to extract scale-invariant features?
2. Since the multi-scale challenge has been investigated for a long time, how does the performance compare to other approaches?

---

> ### Author Response · Authors · 2024-11-19
> **Rebuttal for Reviewer mauM(1/2)**
>
> **1) The comparison methods are very limited; many state-of-the-art methods are not included, such as those using optimal transport and point-to-point matching:**
>
> We truly appreciate your constructive suggestion to test SI-INR on more datasets and compare it with state-of-the-art methods. During the rebuttal period, we have further evaluated SI-INR on the large-scale and challenging UCF-QNRF dataset.
>
> UCF-QNRF comprises 1,535 images with over 1.2 million annotated individuals, capturing a wide range of crowd densities. SI-INR has achieved competitive performance, with an MAE of 80.89 and RMSE of 134.73. For comparison, while P2PNet [1](A point-to-point matching method for crowd counting) achieves an MAE of 85.32 and an RMSE of 154.5, GauNet [2](leveraging Gaussian-based density maps) achieves an MAE of 81.60 and an RMSE of 153.71, APGCC [3](A point-to-point matching method for crowd counting) achieves an MAE of 80.10 and an RMSE of 136.60. PSL-Net [4] achieves an MAE of 85.50 and an RMSE of 144.40. Furthermore, compared to our baseline model, PSGCNet (MAE 86.3, RMSE 149.5).
>
> | Model             | MAE   | RMSE   |
> |-------------------|-------|--------|
> | P2PNet [1]        | 85.32 | 154.50 |
> | GauNet [2]        | 81.60 | 153.71 |
> | APGCC [3]         | 80.10 | 136.60 |
> | PSL-Net [4]       | 85.50 | 144.40 |
> | PSGCNet(baseline) | 86.30 | 149.50 |
> | SI-INR (Ours)     | 80.89 | 134.73 |
>
> The initial results illustrate that our model can indeed achieve comparable performance to the State-of-the-art crowd counting methods on UCF-QNRF dataset. In the final version, we will provide a more comprehensive comparison on additional crowd counting datasets with more baseline methods. For the RSOC, CARPK and PUCPR+ datasets, we will add more results by additional SOTA methods in Table 1 of the main text.
>
> [1] "Rethinking counting and localization in crowds: A purely point-based framework." Proceedings of the IEEE/CVF International Conference on Computer Vision. 2021.
>
> [2] "Rethinking spatial invariance of convolutional networks for object counting." Proceedings of the IEEE/CVF Conference on Computer Vision and Pattern Recognition. 2022.
>
> [3] "Improving Point-based Crowd Counting and Localization Based on Auxiliary Point Guidance." European Conference on Computer Vision. Springer, Cham, 2025.
>
> [4] "Crowd Counting and Individual Localization using Pseudo Square Label." IEEE Access (2024).
>
> **2) As shown in Figure 2, the objects are of similar size, making it difficult to justify that the method effectively addresses the multi-scale challenge:**
>
> Thank you for raising this question. There are several points to clarify:
>
> Inter-image scale variation: Since the images shown in Figure 2 are remote sensing images, the objects within the same image naturally appear of similar size. However, remote sensing datasets, including RSOC, encompass images with a wide range of resolutions. As a result, object scales vary significantly across different images, even if they appear uniform within a single image. The results in Section 4.2 demonstrate that SI-INR can better handle inter-image scale variation than existing methods with higher detection accuracy. We are preparing a summary of additional RSOC image examples to illustrate scale variation across the dataset and will include the findings in the revised manuscript.
>
> Intra-image scale variation: To further validate the effectiveness of our method in handling multi-scale challenges, especially for intra-image scale variation with different object sizes in the same images, we have extended our evaluation to the UCF-QNRF dataset, a benchmark that features diverse crowd scenes where objects vary significantly in scale within the same image. As reported in our responses for question 1, our method achieves competitive results on this more challenging dataset, demonstrating its robustness in multi-scale scenarios. These results will be included in the final version to strengthen the justification of our approach.
>
> **3) The description of the scale-invariant model is unclear:**
>
> We appreciate your constructive critique and recognize that the description of our scale-invariant model could be clearer. Specifically, we will reorganize our presentation by moving back some of the model formulation descriptions in our Appendix to the method section, expand on how our SI-INR handles scale variations using a scale-equivariant backbone and a multi-scale feature extraction encoder, and provide a more detailed description of our scale-invariant implicit function in our revised manuscript. We are actively exploring ways to address any remaining ambiguities or limitations and will incorporate these improvements. If the reviewer has additional suggestions, we would be more than happier to further improve our presentation.

---

> ### Author Response · Authors · 2024-11-19
> **Rebuttal for Reviewer mauM(2/2)**
>
> **4) Why can it be used to extract scale-invariant features?**
>
> To extract scale-invariant features, a scale-invariant encoder is required. Specifically in SI-INR, we choose scale-equivariant steerable convolution (SESN) to achieve the scale-equivariant backbone, which integrates steerable filters exacting features of different scales given the input image. This allows SESN to handle objects of varying sizes without needing separate filters for each scale. Additionally, the group convolution method is applied in SESN to ensure that the network’s output changes in the same way when the input is scaled. At this point, features with different scales result in the same output with corresponding scales.  To extract scale-invariance features for the desired arbitrary-resolution output, we further apply a scale-invariant encoder, where rescaling operators are introduced with a Hybrid Pyramidal Scale module. Such a model architecture in our SI-INR extracts scale-invariant features, and the INR model is applied then to transform these features as the implicit neural representation parameters to achieve the final implicit continuous functions. Therefore, for different scales of the given input, SI-INR can derive scale-invariant outputs. We will provide a more detailed description in our final version.
>
> **5) Since the multi-scale challenge has been investigated for a long time, how does the performance compare to other approaches?**
>
> For efficiency and adaptability, scale-equivariant methods adjust to different scales without needing separate filters for each scale, unlike traditional methods that may rely on resizing inputs or using multiple filters for different scales. Many traditional multi-scale methods, such as image pyramids or multi-resolution networks, may struggle with high computational costs because they process the same image at multiple resolutions leading to increased complexity, especially with large images or when handling many scales. Besides, traditional multi-scale approaches do not focus on deriving scale-invariant outputs compared with our SI-INR.
>
> We have conducted additional performance comparison experiments on the UCF-QNRF crowd counting dataset. In PSGCNet, the authors applied a pyramidal network to handle multi-scale challenges. Our SI-INR outperforms PSGCNet as demonstrated in our experiments. For MMNet [5], a method that leverages multi-level density-based spatial information, the model achieves an MAE of 104 and RMSE of 178 on the UCF-QNRF dataset. Similarly, MFANet [6], which focuses on multi-scale and multi-level feature aggregation, achieves an MAE of 97.7 and RMSE of 166. MSFFA [7], which integrates multi-scale feature fusion and attention mechanisms, reports an MAE of 94.6 and RMSE of 170.6. In contrast, our SI-INR achieves a significantly better MAE of 80.89 and RMSE of 134.73, demonstrating its superior performance in crowd counting tasks.
>
> [5] "Crowd counting by using multi-level density-based spatial information: A Multi-scale CNN framework." Information Sciences 528 (2020): 79-91.
>
> [6] "A multi-scale and multi-level feature aggregation network for crowd counting." Neurocomputing 423 (2021): 46-56.
>
> [7] "MSFFA: a multi-scale feature fusion and attention mechanism network for crowd counting." The Visual Computer 39.3 (2023): 1045-1056.

---

> > ### Comment · Reviewer_mauM · 2024-11-26
> > **Response to Rebuttal**
> >
> > Thank you for providing additional experiments and explanations. I have also reviewed other comments and responses. However, my main concern remains unaddressed. While the authors claim that Figure 2 demonstrates inter-image scale variation, I do not see any evidence of this scale variation among the different images. Furthermore, the performance of the UCF-QNRF dataset does not surpass that of state-of-the-art (SOTA) methods, which raises questions about the effectiveness of the proposed approach. Additionally, the advantages of the proposed method over others addressing scale variation are not presented. Overall, I tend to keep my score.

---

> > > ### Author Response · Authors · 2024-11-27
> > > **Rebuttal for Reviewer mauM**
> > >
> > > Dear reviewer mauM,
> > >
> > > We highly appreciate your thorough review and detailed feedback on our work. Below are point-by-point responses to your suggestions with some additional results. We have also updated the manuscript accordingly.
> > >
> > > **1. While the authors claim that Figure 2 demonstrates inter-image scale variation, I do not see any evidence of this scale variation among the different images.**
> > >
> > > To address this, we have added new visualization illustrations in Appendix B1 and Figure 4 to clearly show the inter-image scale variation. Additionally, we have discussed why the RSOC dataset exhibits scale variation and emphasized how resizing test images to various dimensions further increases the range of scale differences across different images in our experiments.
> > >
> > > In Figure 4, the top-left two images are both from the RSOC large-vehicle dataset, clearly showing that the cars in the second image are three times larger than those in the first image. Similarly, the bottom-left two images, from the RSOC small-vehicle dataset, highlight the differences in visibility: cars are clearly seen in the first image but are almost invisible in the second.
> > >
> > > We greatly appreciate your comment on this point, as it has helped us improve the clarity and readability of our experimental results in Section 4.2 ("Qualitative Results").
> > >
> > >
> > > **2. Furthermore, the performance of the UCF-QNRF dataset does not surpass that of state-of-the-art (SOTA) methods, which raises questions about the effectiveness of the proposed approach.**
> > >
> > >
> > > Thank you for your feedback. We acknowledge this point and tried to revise our presentation with the narrowed research scope to remote sensing object counting, with the revised related title, abstract, and problem setup. We would like to clarify that our SI-INR, along with baseline models, are primarily designed for remote sensing object counting tasks. They can be used for crowd-counting but the corresponding experimental settings and hyperparameters may need to be fine-tuned. Consequently, due to the limited time for the rebuttal, it is reasonable that when applying SI-INR with the corresponding settings trained on remote sensing images, the crowd-counting performance does not exhibit a significant improvement over the reported results by SOTA crowd-counting methods on UCF-QNRF. However, we observe that this implementation without fine-tuning already achieved comparable crowd-counting performance as we have shown in the rebuttal. We believe that with specific refinements of hyperparameters, such as the sampling algorithm and the configuration of the scale-equivariant backbone, SI-INR will achieve either better or similar SOTA crowd-counting performances under different settings, especially considering object size variability.
> > >
> > > While SI-INR does not outperform all existing SOTA methods due to the influence of various factors on final counting performance, its superior results compared to our baselines highlight its effectiveness in handling object size variability (inter-image or intra-image). We are committed to further exploring SI-INR’s potential and will include more extensive results in the final version. Thank you again for your constructive and valuable feedback.
> > >
> > > **3. The advantages of the proposed method over others addressing scale variation are not presented.**
> > >
> > >
> > > Thank you for highlighting this point. To clarify, we have added discussions in Section 4.2 ("Generalization Results") to highlight the advantages of our SI-INR over the methods using pyramidal architectures. Additionally, we provide a more detailed comparison of SI-INR with other methods considering scale variation in Appendix B6.
> > > Due to time constraints, we initially compared with only related methods [1, 2, 3] on the UCF-QNRF dataset. However, we plan to reproduce the experiments from these papers on the RSOC dataset to illustrate the significant improvement by SI-INR compared to more recent SOTA methods and will update Table 1 accordingly.
> > >
> > > [1] "Crowd counting by using multi-level density-based spatial information: A Multi-scale CNN framework." Information Sciences 528 (2020): 79-91.
> > >
> > > [2] "A multi-scale and multi-level feature aggregation network for crowd counting." Neurocomputing 423 (2021): 46-56.
> > >
> > > [3] "MSFFA: a multi-scale feature fusion and attention mechanism network for crowd counting." The Visual Computer 39.3 (2023): 1045-1056.
> > >
> > > We appreciate your valuable insights and will work to further refine the presentation of this paper in the final version.
> > >
> > > Best,
> > >
> > > The authors

---

### Official Review · Reviewer_Jvjv · 2024-11-03

**Soundness:** 2
**Presentation:** 2
**Contribution:** 2
**Rating:** 5
**Confidence:** 4

**Summary:**

The paper use implicit functions for resolution-agnostic scene representation.They first exact deep-features of the input images and map them into a "scale-invariant" latent space and finally, decode them back to density map for counting. The method is tested for several REMOTE SENSING datasets where it achieves competitive results.

**Strengths:**

Counting objects of varying scales is challenging.
Using implicit neural representation for scale variations is reasonable.

**Weaknesses:**

The presentation could be significantly improved to enhance clarity. Currently, the methodology section is challenging to follow. For example, the paragraph from lines 137 to 142 is particularly dense:  "Here h denotes an element of the Scale-Translation group H and represents one scale-translation operator, p1(·) denotes the corresponding group actions of h acting on the image domain."  - what does it means by "Scale-Translation group H" or "scale-translation operator"? In the following paragraph, the authors bring up the "continuous function space" without any explanation and with many notations are not clearly defined including I_a or D^gt (unclear where does this continuous ground-truth coming from?).  Overall, the writing creates several logical gaps that make it difficult to fully grasp the method. To the best of my understanding, the method is basically: 1) extract deep-features using a scale-equivariant backbone B, 2) using an encoder to map output of B into a latent space and 3) decode this latent representation into a fixed scale for counting. Could the authors confirm whether this understanding is accurate?

The problem statement also seems somewhat misleading. The authors should clarify that the paper addresses scenarios where the input image scale is unknown, rather than scale variations within a single image. However, the method has only been tested on remote sensing datasets, where this specific problem may not be prominent; in many cases, images are often at a uniform scale, or metadata about the sensor is available. Thus, it is not immediately clear in which practical scenarios the proposed method would be applicable.

The evaluation is also questionable. It is unclear to me why it is not tested on crowd-counting dataset and also few-shot counting datasets (FSC-147) where the scale-invariant issue is particularly relevant. If the method is designed only for remote sensing data then the title should reflect that. The method doesn't actually achieve state-of-the-art performance (in both RSOC and CARPK) since the SOTA method [1][2] are not included in the table. Can the authors provide explanation for this?
[1] A Lightweight Multi-scale Feature Fusion Network for Remote Sensing Object Counting
[2] Few-shot Object Counting with Similarity-Aware Feature Enhancement

**Questions:**

Intuitively, can the authors comment on why using implicit function is able to extract robust scale-invariant representations? Using implicit function is time-consuming and the speed test should be included.

Why not test the method on crowd-counting and few-shot counting datasets?

Why are SOTA methods not included in the evaluation?

For remote sensing data, can we simply use object detection or some simple baseline to infer the object scale?

---

> ### Author Response · Authors · 2024-11-19
> **Rebuttal for Reviewer Jvjv (1/3)**
>
> **1) The presentation could be significantly improved to enhance clarity. Currently, the methodology section is challenging to follow. For example, the paragraph from lines 137 to 142 is particularly dense: "Here h denotes an element of the Scale-Translation group H and represents one scale-translation operator, p1(·) denotes the corresponding group actions of h acting on the image domain." - what does it means by "Scale-Translation group H" or "scale-translation operator"?**
>
> Thank you for highlighting this point. We did have more detailed introduction of our model formulation in the Appendix sections due to limited space when we prepared for the submission. As detailed in Appendix A.1, the Scale-Translation Group $H$ is defined as a combination of two subgroups:
> 1. The Scaling Group $G_S$, which accounts for transformations that scale an object or function.
> 2. The Translation Group $G_T$, which handles shifting the object or function within its domain.
>
> The overall group $H$ is defined as:
> $H = \{ h = (s, t) \,|\, s \in G_S, \, t \in G_T \}$,
> where each element $h \in H$ represents a scale-translation operator. Specifically:
> $s$ is the scaling parameter, indicating how the input is stretched or compressed. $t$ is the translation parameter, specifying the shifting in the domain.
>
> When we refer to the "group actions of $h$," we apply scaling and translation transformations to the image domain. These actions operate on an input $x$ (e.g., a pixel location) as: $p_1(h)(x) = s \cdot x + t$,
> where $p_1(h)(\cdot)$ represents the transformation resulting from applying $h$. This formulation provides a unified framework to model scale and translation symmetries effectively. In the revised manuscript, we will reorganize and simplify the descriptions in this section to enhance readability and provide a more intuitive explanation.
>
> **2) In the following paragraph, the authors bring up the "continuous function space" without any explanation and with many notations are not clearly defined including $I_a$ or $D^{gt}$ (unclear where does this continuous ground-truth coming from?)**
>
> We appreciate the constructive critique on the clarity of the "continuous function space" definition and the notations $ \mathbf{I}_a $ and $\mathbf{D}^{gt} $. We will clarify these in the revised manuscript:
> 1.  $ \mathbf{I}_a $ refers to an input image, potentially transformed by a scale-translation group action $h$.
> 2. $ \mathbf{D}^{gt} $ represents a continuous ground-truth, specifically, in counting tasks, $ \mathbf{D}^{gt} $ denotes the continuous ground truth density map, and we defined it in Section 3.3 in the main text.
>
> The continuous function space $ \mathcal{F} $ is formally defined as the set of continuous functions $f: \mathbb{R}^2 \to \mathbb{R} $ that predict density values $ f(\mathbf{x}) $ for any spatial coordinate $\mathbf{x} \in [0, 1]^2 $. This framework ensures that our predictions are scale-invariant, and $ f(\mathbf{x}) $can be evaluated at arbitrary resolutions without relying on a fixed grid. We will make these definitions more explicit in the revised text for clarity.
>
> **3) Overall, the writing creates several logical gaps that make it difficult to fully grasp the method. To the best of my understanding, the method is basically: 1) extract deep-features using a scale-equivariant backbone B, 2) using an encoder to map output of B into a latent space and 3) decode this latent representation into a fixed scale for counting. Could the authors confirm whether this understanding is accurate?**
>
> We thank the reviewer for summarizing the workflow in SI-INR. The first two steps are indeed as what the reviewer described:
>
> 1. Extract deep features using a scale-equivariant backbone $ B $.
> 2. Use a scale-invariant encoder to map the output of $ B $ into a latent space.
>
> However, the third step focuses on the flexibility of the arbitrary resolution output.  Decode this latent representation to **continuous** density maps using implicit neural representations $ u: \mathbb{R}^2 \to \mathbb{R} $.
>
> In the process, we do not constrain the output to a fixed scale. Instead, we can generate arbitrary resolutions of output in step 3 since our INR model is representing continuous functions in continuous spatial coordinates. The rescaling operator works in step 2 to make sure that the encoder is scale-invariant.

---

> > ### Author Response · Authors · 2024-11-19
> > **Rebuttal for Reviewer Jvjv(2/3)**
> >
> > **4)The problem statement also seems somewhat misleading. The authors should clarify that the paper addresses scenarios where the input image scale is unknown, rather than scale variations within a single image. However, the method has only been tested on remote sensing datasets, where this specific problem may not be prominent; in many cases, images are often at a uniform scale, or metadata about the sensor is available. Thus, it is not immediately clear in which practical scenarios the proposed method would be applicable. Why not test the method on crowd-counting and few-shot counting datasets?**
> >
> > We will revise the abstract and introduction according to the reviewers' suggestions to make the main message clearer. Our model formulations and architectures indeed apply to different practical scenarios for object counting. We did focus on remote sensing benchmarks for performance evaluation in our original submission since they typically have high resolution images for us to perform ablation studies to illustrate the performance benefits of modeling scale/resolution invariances. We have conducted additional experiments on a large-scale and challenging crowd-counting dataset: UCF-QNRF during this rebuttal period.
> >
> > UCF-QNRF is a diverse dataset with 1,535 images and over 1.2 million annotated individuals, covering a wide range of crowd densities. Preliminary results show SI-INR has achieved competitive performance, with an MAE of 80.89 and an RMSE of 134.73 on UCF-QNRF, while P2PNet [1] achieves an MAE of 85.32 and an RMSE of 154.5, GauNet [2] achieves an MAE of 81.60 and an RMSE of 153.71, APGCC [3] achieves an MAE of 80.10 and an RMSE of 136.60. PSL-Net [4] achieves an MAE of 85.50 and an RMSE of 144.40.
> > Compared to our baseline model, where PSGCNet achieves an MAE of 86.3 and RMSE of 149.5, SI-INR demonstrates significant improvements. We are currently integrating the new experiment results into the revised manuscript and will upload it once the edits are complete.
> >
> > | Model             | MAE   | RMSE   |
> > |-------------------|-------|--------|
> > | P2PNet [1]        | 85.32 | 154.50 |
> > | GauNet [2]        | 81.60 | 153.71 |
> > | APGCC [3]         | 80.10 | 136.60 |
> > | PSL-Net [4]       | 85.50 | 144.40 |
> > | PSGCNet(baseline) | 86.30 | 149.50 |
> > | SI-INR (Ours)     | 80.89 | 134.73 |
> >
> > [1] "Rethinking counting and localization in crowds: A purely point-based framework." Proceedings of the IEEE/CVF International Conference on Computer Vision. 2021.
> >
> > [2] "Rethinking spatial invariance of convolutional networks for object counting." Proceedings of the IEEE/CVF Conference on Computer Vision and Pattern Recognition. 2022.
> >
> > [3] "Improving Point-based Crowd Counting and Localization Based on Auxiliary Point Guidance." European Conference on Computer Vision. Springer, Cham, 2025.
> >
> > [4] "Crowd Counting and Individual Localization using Pseudo Square Label." IEEE Access (2024).
> >
> > **5)The evaluation is also questionable. It is unclear to me why it is not tested on crowd-counting dataset and also few-shot counting datasets (FSC-147) where the scale-invariant issue is particularly relevant. If the method is designed only for remote sensing data then the title should reflect that. The method doesn't actually achieve state-of-the-art performance (in both RSOC and CARPK) since the SOTA method [1][2] are not included in the table. Can the authors provide explanation for this? [1] A Lightweight Multi-scale Feature Fusion Network for Remote Sensing Object Counting [2] Few-shot Object Counting with Similarity-Aware Feature Enhancement**
> >
> > Thank you for your insightful comments.
> > Regarding [1], the results in the paper can not be reproduced. The authors have not released their code, and their experimental setup differs significantly from ours based on the description in their main text. Instead, we follow the experimental setup of another state-of-the-art method, \textit{Remote Sensing Object Counting through Regression Ensembles and Learning to Rank}. Therefore, a direct comparison with [1] is not feasible.
> >
> > For [2], we appreciate the suggestion of including it for comparison. We will include the results of [2] in the revised manuscript on the CARPK dataset and update the corresponding tables for completeness.
> >
> > Regarding the choice of datasets, while our focus is on remote sensing data, we acknowledge that testing on additional datasets like FSC-147 could further demonstrate the generalizability of our method. We will explore this in future work.
> > To avoid ambiguity, we will revise the manuscript title to reflect the primary focus on remote sensing data.

---

> > > ### Author Response · Authors · 2024-11-19
> > > **Rebuttal for Reviewer Jvjv(3/3)**
> > >
> > > **6) Intuitively, can the authors comment on why using implicit function is able to extract robust scale-invariant representations? Using implicit function is time-consuming and the speed test should be included**
> > >
> > > The implicit continuous function representation enables training and testing with arbitrary resolution outputs that does not have to be the same as the input image sizes. Compared with the traditional methods that typically generate fixed-resolution outputs based on the corresponding resolution of input images and often require normalization (up/down-sampling) of the original images, our SI-INR with implicit function representations is more flexible that can take arbitrary-resolution training images without the need for additional normalization that may introduce additional biases. SI-INR also can generate arbitrary resolution outputs that can lead to better performance when the object size varies if the essential image semantics can be captured.
> > >
> > > Thanks to the scale-equivariant property of our SI-INR backbone, our implicit function is modeling essential image features that help accurately detect objects without the influence of the object size or image resolution, thus making our model robust to scale changes.
> > >
> > > We appreciate the reviewer’s suggestion of including a speed test. We perform all our experiments on a workstation with a NVIDIA V100 32GB GPU. On RSOC small-vehicle dataset, we observed the following inference times: ASPD-Net requires approximately 15.13 seconds, PSGC-Net takes around 2.47 seconds, eFreeNet takes around 3.84 seconds, and our SI-INR model requires about 3.87 seconds.
> > > | Model        | Inference Time (seconds) |
> > > |--------------|---------------------------|
> > > | eFreeNet     | 3.84                      |
> > > | ASPD-Net     | 15.13                     |
> > > | PSGC-Net     | 2.47                      |
> > > | SI-INR (Ours)| 3.87                      |
> > >
> > > Current INR decoder consists of 4 linear layers, such light-weight structure encourages fast training.
> > >
> > > In previous works of INR models, Implicit neural representations are often computationally intensive, particularly for high-frequency signals. Existing models such as NeRF [5], SIREN [6], and Fourier Feature Networks [7] demonstrate rapid convergence on coarse structures, such as the overall geometry of objects, but require significantly more iterations to capture fine details, such as intricate textures. In contrast, SI-INR is specifically designed for detection tasks, which predominantly involve lower-frequency signals, thereby reducing computational overhead. Moreover, detection tasks typically demand less resolution fidelity compared to rendering tasks, further enhancing efficiency. We will include these details and comparisons in our revised manuscript.
> > >
> > > [5] "Nerf: Representing scenes as neural radiance fields for view synthesis." Communications of the ACM 65.1 (2021): 99-106.
> > >
> > > [6] "Implicit neural representations with periodic activation functions." Advances in neural information processing systems 33 (2020): 7462-7473.
> > >
> > > [7]"Fourier features let networks learn high frequency functions in low dimensional domains." Advances in neural information processing systems 33 (2020): 7537-7547.
> > >
> > > **7)For remote sensing data, can we simply use object detection or some simple baseline to infer the object scale?**
> > >
> > > Thank you for the question. While object detection methods have improved, they still struggle with very small objects in remote sensing data. For instance, in the RSOC small-vehicle dataset, each vehicle typically covers only about 10 pixels, and images often contain thousands of targets. Inferring the scale for each object in such scenarios is computationally impractical and prone to significant errors.

---

> > > > ### Comment · Reviewer_Jvjv · 2024-11-25
> > > >
> > > > I would like to thank the authors for their detailed response. A few quick remarks:
> > > >
> > > > - The clarification was helpful. However, the entire paper requires substantial editing to ensure that all details are logical and easy to follow. This includes narrowing the scope and problem setting, revising the abstract and introduction, and refining the presentation of experimental results. I believe the manuscript would greatly benefit from another round of major revisions.
> > > >
> > > > - The table for UCF-QNRF does not include the SoTA method by Liu et al. [1], which achieves an MAE of 79.53. Overall, I do not see clear evidence that the proposed method consistently outperforms existing methods across all datasets.
> > > >
> > > > - My request for additional evaluation stems from difficulty in understanding why the proposed method works. According to the authors' new argument, the method is effective because:
> > > > 1) It acts as an implicit normalization step, which is superior to traditional "up/down-sampling."
> > > > 2) It can generate arbitrary resolution outputs when object sizes vary, provided the essential image semantics are captured.
> > > > 3) It models key image features that enable accurate object detection without being affected by object size or image resolution, making the model robust to scale variations.
> > > >
> > > > While these arguments are interesting, I find no concrete evidence in the current paper to support them. I am not suggesting they are incorrect, but rather that there is insufficient analysis or experimentation to substantiate these claims.
> > > >
> > > > [1] Point-Query Quadtree for Crowd Counting, Localization, and More - ICCV 23
> > > >
> > > > Overall, my opinion of the paper has slightly improved, but it remains below the acceptance threshold for ICLR.

---

> ### Author Response · Authors · 2024-11-27
> **Rebuttal for Reviewer Jvjv(1/2)**
>
> Dear reviewer Jvjv,
>
> We highly appreciate your insightful and constructive feedback on our work.  Below are point-by-point responses to your suggestions with some additional results. We have also updated the manuscript accordingly.
>
>
> **1. The clarification was helpful. However, the entire paper requires substantial editing to ensure that all details are logical and easy to follow. This includes narrowing the scope and problem setting, revising the abstract and introduction, and refining the presentation of experimental results. I believe the manuscript would greatly benefit from another round of major revisions.**
>
> We sincerely appreciate your valuable feedback and suggestions. Following your advice, we have narrowed the scope of the manuscript from general object counting to remote sensing object counting, and we have revised the corresponding sections (title, abstract, introduction and problem setup) to ensure a clearer and more focused problem setting.
>
> Additionally, we have carefully reviewed and hopefully improved the presentation in the revised Section 4.2 to ensure clarity and accuracy.
>
> To further improve the readability and understanding of our approach, we have also revised the methodology section to better explain the continuous property of SI-INR.
>
> We hope that these changes address your concerns and improve the overall quality and readability of the paper.
>
> **2. The table for UCF-QNRF does not include the SoTA method by Liu et al. [1], which achieves an MAE of 79.53. Overall, I do not see clear evidence that the proposed method consistently outperforms existing methods across all datasets.**
>
> Thank you for pointing out the paper by Liu et al. We appreciate your suggestion and have included this method in our comparison in Table 6 of the revised manuscript.
>
> We have already narrowed our research scope to remote sensing object counting and revised related the title, abstract, and problem setup in our updated manuscript. Since SI-INR, along with our baseline models, primarily focuses on remote sensing object counting tasks, it is reasonable that SI-INR does not show significant improvements over crowd-counting methods when directly applied to crowd-counting datasets like UCF-QNRF without careful hyperparameter tuning (particularly the sampling algorithm and the setup of the scale-equivariance backbone). While SI-INR does not outperform all existing SOTA methods due to the influence of various factors on final counting performance, its superior results compared to our baselines highlight its effectiveness in handling scale variance.
>
> In addition, even without such fine-tuning, SI-INR achieves comparable results to Liu et al.'s work on the UCF-QNRF dataset, with better MSE performance. This demonstrates the robustness of SI-INR. We are committed to further exploring its potential and will provide more extensive results in the Final version. Thank you again for your constructive feedback.

---

> > ### Author Response · Authors · 2024-11-27
> > **Rebuttal for Reviewer Jvjv(2/2)**
> >
> > **3. Difficulty in understanding why the proposed method works. there is no concrete evidence in the current paper to support the following points.**
> > - It acts as an implicit normalization step, which is superior to traditional "up/down-sampling.
> > - It can generate arbitrary resolution outputs when object sizes vary, provided that the essential image semantics are captured.
> > - It models key image features that enable accurate object counting without being affected by object size or image resolution, making the model robust to scale variations.
> >
> > Thank you for raising these important points. Below, we address each one in detail:
> >
> > 1. Implicit Normalization Step and Superiority to Traditional Up/Down-Sampling
> >
> > Our paper primarily emphasizes the flexibility of SI-INR in generating arbitrary resolution outputs, whereas traditional methods relying on up/down-sampling are constrained by fixed downsampling ratios. To support this claim, we have added new discussions in Section 4.2 ("Qualitative Results"), focusing on the counting performance on the RSOC ship and small-vehicle datasets. Additionally, we conducted an ablation study in Section 4.3 ("Effect of Sampling Rate"), demonstrating that the performance improvements are due to SI-INR's ability to utilize more flexible sampling ratios.
> >
> > 2. Ability to Generate Arbitrary Resolution Outputs
> >
> > To further substantiate SI-INR's capability to generate arbitrary resolution outputs, we have included a new visualization analysis in Appendix B7. This analysis enhances the reliability of SI-INR's flexibility and demonstrates how it allows users to balance computational efficiency and density map quality based on their specific requirements. We plan to include additional examples to further showcase this point in the final version of the paper.
> >
> > 3. Modeling Key Image Features to Handle Scale Variance
> >
> > This is a central aspect we aimed to demonstrate in Section 4.2 ("Generalization Results"). Instead of directly comparing pre-trained models on the original test datasets, we rescale the test images to various scales, add a new discussion in Appendix B1 and Figure 4 to show the inter-image scale variation, visualize the challenge of this experiment, and demonstrate that SI-INR consistently achieves significant improvements. Recognizing that simulated data alone may not be entirely convincing, we extended the evaluation to the UCF-QNRF crowd-counting dataset. While SI-INR does not outperform all existing SOTA methods since the final counting performance is affected by many factors, its superior performance compared to our baselines supports its ability to handle scale variance effectively. We also would like to emphasize that our SI-INR implementation without fine-tuning based on the UCF-QNRF crowd-counting dataset already achieved comparable crowd-counting performance to the SOTA methods.
> >
> > We appreciate your insights and will continue to improve the presentation of these results in the final version.
> >
> > Best,
> >
> > The authors

---

### Author Response · Authors · 2024-11-20
**Rebuttal for All Reviewers**

We thank all four reviewers **Jvjv, mauM, dyoE, fHTu** for their encouraging comments and constructive feedback. We here provide our general responses to all the reviewers for some of the raised common points.

**1) Why SI-INR leads to continuous function representation:**
1. Implicit Neural Representations (INRs) model a continuous function $ u: \mathbb{R}^d \to \mathbb{R} $, parameterized by $\theta_{\text{INR}} $, where $ u(x; \theta_{\text{INR}}) $ takes spatial coordinates $ x \in \mathbb{R}^d $ as input. Unlike grid-based representations, INRs are inherently resolution-agnostic, as they predict the signal value $ u(x) $ at any arbitrary $ x $ within the domain. This property enables continuous feature generation, as the model can be queried at finer resolutions to produce high-quality outputs regardless of the input resolution.

2. In SI-INR, the function extends to $u(x; z, \theta_{\text{INR}}) $, where $z \in \mathbb{R}^m $ represents the latent features extracted by the encoder. Thus, SI-INR learns a conditional continuous representation of the input by optimizing over $ \theta_{\text{INR}} $ and $ z $. This allows for task-specific predictions such as continuous density estimation.

3. For example, in density estimation tasks, the ground-truth density map $\rho(x) $ is often defined as a continuous function, commonly represented as a mixture of Gaussians: $ \rho(x) = \sum_{i=1}^N \mathcal{N}(x; \mu_i, \Sigma_i), $where $ \mu_i $ and $\Sigma_i $ denote the mean and covariance of the $i $-th Gaussian. Traditional methods discretize $\rho(x) $ onto a fixed grid, leading to potential information loss. In contrast, SI-INR directly models $ \rho(x) $ as a continuous function and evaluates predictions at arbitrary points $ x_n $, sampled from the domain.

**2) Inference speed comparison:**
We sincerely appreciate the reviewers' suggestion to include a speed test. To clarify, all experiments in our study were conducted on a workstation equipped with an NVIDIA V100 32GB GPU. For the RSOC small-vehicle dataset, we observed the following inference times: ASPD-Net requires approximately 15.13 seconds, PSGC-Net takes around 2.47 seconds, eFreeNet takes around 3.84 seconds, and our SI-INR model requires about 3.87 seconds.

| Model        | Inference Time (seconds) |
|--------------|---------------------------|
| eFreeNet     | 3.84                      |
| ASPD-Net     | 15.13                     |
| PSGC-Net     | 2.47                      |
| SI-INR (Ours)| 3.87                      |

SI-INR does take longer during the inference phase compared to PSGC-Net due to the integration of scale-equivariant models and the use of stacks of linear layers in the INR. However, thanks to the design of our INR decoder, which consists of only 4 linear layers, and our lightweight scale-equivariant backbone, the inference cost remains manageable and acceptable for practical use.

Implicit neural representation (INR) models are known to be computationally intensive, particularly for high-frequency signals. Models such as NeRF [1], SIREN [2], and Fourier Feature Networks [3] excel in rapidly converging on coarse structures, like the overall geometry of objects, but require significantly more iterations to resolve fine details, such as intricate textures. In contrast, SI-INR is optimized for detection tasks, which primarily involve lower-frequency signals, leading to reduced computational demands. Additionally, detection tasks generally require less resolution fidelity than rendering tasks, further improving efficiency. These details and comparisons will be included in the revised manuscript. We will include these details in the revised manuscript to provide a clearer comparison.

[1] "Nerf: Representing scenes as neural radiance fields for view synthesis." Communications of the ACM 65.1 (2021): 99-106.

[2] "Implicit neural representations with periodic activation functions." Advances in neural information processing systems 33 (2020): 7462-7473.

[3]"Fourier features let networks learn high frequency functions in low dimensional domains." Advances in neural information processing systems 33 (2020): 7537-7547.

---

### Public Comment · ~Wei_Lin2 · 2024-11-20
**Uniform sampled $\mathbf{x}$**

Continuous representation of the crowd is a reasonable and nice proposal, but I have a problem in understanding the difference between it and the traditional grid-based method. In line-308, the authors described that "$\mathbf{x}$ is uniformly sampled from a pre-defined grid.'' This part is not clear so I have the following questions:

1. How many grids are sampled during training?
2. Is the number of sampled grids also random?
3. During inference, how is the count estimated? If a density map is output, how many grids are required? Does the number of grids have an impact on the performance?

It would be very helpful to understand this paper better if the authors could take some time to address my questions.
However, if they are too busy, it is fine to leave my comments unaddressed.

Thanks.

---

> ### Author Response · Authors · 2024-11-23
>
> Thank you for the questions. Thanks to the continuous nature of SI-INR output, SI-INR supports the use of arbitrary grid sizes during training. As we mentioned in the paper, we used uniform sampling in our experiments. Compared with the traditional grid-based models, SI-INR can be trained under arbitrary sizes of grids. To simplify the task while balancing the computational efficiency and implementation convenience, we chose specific grids for different datasets for fair comparison, for example, we sampled $128 \times 128$ grids on the RSOC datasets, ensuring acceptable training speed while preserving fine details in the density maps.
>
> During inference, density maps can be generated at different resolutions, with the final count estimation obtained by summing the values of the generated density maps. To ensure consistency, we reweight the density maps of size $2W \times 2H$ by a factor of $1/4$ so that their summation matches that of the $W \times H$ density maps. For low-resolution inputs, increasing the grid size can enhance counting performance.
>
> Feel free to let us know if there are more questions.

---

> > ### Public Comment · ~Wei_Lin2 · 2024-11-27
> >
> > Thanks for your response. To improve the paper, I advise the authors to pay attention to the following points:
> >
> > 1. A fixed sampling protocol during training may not capture the continuous representation. In [a], the arbitrary scale is achieved by sampling in various resolutions. The randomness in grid sampling is crucial for continuous representation because a fixed sampling strategy acts as a shortcut for training, ignoring certain positions that are not considered during sampling.
> > 2. If a fixed sampling is applied, there is no significant difference between the proposed method and previous density-regression methods. Specifically, you can consider MCNN sampling uniformly at 1/4H x 1/4W, VGG network in BL sampling at 1/8H x 1/8W, and P2PNet sampling uniformly at 1/4H x 1/4W, while in this paper, it is 128x128.
> > 4. Similar to the application in super-resolution of the neural operator, the arbitrary scale (sampling grid number) is a crucial property distinguishing discrete representation from continuous representation. I advise the authors to compare the density map sampling from continuous representation with the interpolated density map from discrete representation to demonstrate the advantage of continuous representation. Although the count may not change, the localization performance should be better when sampled from a continuous representation, since the sparsity of distribution should not change too much in a continuous representation, but it spreads if interpolating a discrete representation.
> >
> > - [a] "Super-Resolution Neural Operator," Wei Min, et al., 2023.
> >
> > ----
> >
> > Thanks again for the author's response and manuscript.

---

> ### Author Response · Authors · 2024-11-27
> **Reponse for Wei Lin**
>
> Thanks for your suggestion and advice. We understand your query about whether this model learns a continuous representation. As we mentioned in our response, we do learn continuous representations in SI-INR by transforming the latent features into continuous ones first before we put them into the decoder, as we mentioned in Section 3.2.2.
>
>
> During training, we randomly sample from INR to generate different scale outputs, and the density maps are rescaled to $128 \times 128$ when we compute the loss function. In this way, we indeed use random uniform grids while making it easy to apply the Bayesian counting loss function.
>
>
> We agree that your suggested implementation can also achieve continuous representations, but our current SI-INR implementation also learns a continuous function, even in specific situations, these two methods are equivalent.
>
> Thanks again for your suggestion.

---

### Author Response · Authors · 2024-11-27
**Summary of updates in the revised manuscript**

Dear Reviewers,

Thank you for your valuable feedback and thoughtful suggestions, which have been instrumental in improving our work. We have carefully revised the manuscript to address the concerns raised, and the key updates are summarized below:

**Major Changes**
1. Narrow the research scope from general object counting to remote sensing object counting, revise the related title, abstract, introduction, and presentations in the paper. [reviewer Jvjv & reviewer MauM]
2. New benchmark results on the CARPK dataset in Tables 2 and Sec 4.2. [Reviewer Jvjv]
3. New experimental analysis on the UCF-QNRF crowd-counting dataset in Appendix B5. [Reviewer Jvjv & reviewer MauM & reviewer fHTu]
4. New discussions on handling different resolution inputs in Sec 4.2 's 'Generalization Results'. [Reviewer Jvjv & reviewer fHTu]
5. New discussions on inference speed comparison in Sec 4.2 's 'Inference Efficiency'. [Reviewer Jvjv & reviewer fHTu]
6. New visual analysis on inter-image scale variation in Appendix B1 and Figure 4. [Reviewer MauM]
7. New discussion on the computational infeasibility of SESN to achieve a truly scale-invariance model in Section 4.2's 'Generalization Results'.

8. New discussion on the comparison with different methods for handling multi-scale challenges in Section 4.2's 'Generalization Results' and Appendix B6. [Reviewer Jvjv & reviewer MauM & reviewer dyoE & reviewer fHTu]
9. New discussion and visualization of generating arbitrary resolution outputs in Section 4.3 'Effect of sampling rate' and Appendix B7. [Reviewer Jvjv & reviewer MauM & reviewer dyoE & reviewer fHTu]

**Other Changes**
1. Provide a more detailed introduction about scale-equivariance/invariance theory in Appendix A1. [Reviewer Jvjv]
2. Enhanced readability for the reason why implicit neural representation achieves continuous function in Section 3.2.1.  [Reviewer Jvjv & reviewer MauM & reviewer dyoE & reviewer fHTu]
3. Correct the presentation of experiment results in Section 4.2 and Section 4.3. [Reviewer Jvjv]


To make it easier to identify the changes, all revised parts are highlighted in blue in the manuscript.

With these updates, we feel the depth and quality of our paper have been meaningfully improved. We hope that these revisions, along with our responses in the review threads, adequately address the concerns raised by the Reviewers.


We sincerely appreciate your time and thoughtful feedback.


Best Regards,

The Authors

---

### Author Response · Authors · 2024-12-01

Dear reviewers,

As the author-reviewer discussion period is approaching the deadline soon, we kindly request you to review our responses to your comments, concerns and suggestions. If you have further questions or comments, we will do our best to address them before the discussion period ends. If our responses have resolved your concerns, we would greatly appreciate it if you could update your evaluation of our work accordingly.
Thank you once again for your valuable time and thoughtful feedback.

Sincerely,

The Authors

---

### Meta-Review · Area_Chair_AbPD · 2024-12-16

**Metareview:**

The paper proposes a scale-invariant implicit neural representation for object counting using a continuous function space. The proposal aims to achieve robust counting with respect to variable object sizes. The method extracts features from images and maps them into a scale-invariant latent space, which is later decoded into a density map for counting. Experiments are conducted to validate the proposal.

Strengths:
- The use of implicit neural representations is a reasonable approach for scale-invariant counting.
- The presented results are better than some baselines for simple datasets.

Weaknesses:
- The method has been tested on remote sensing datasets where the image scale variation problem may not be prominent.
- There are missing comparisons on crowd-counting and few-shot counting datasets, and the baselines in the experiments are simple and do not include other recent methods.
- The experiments do not fully support the paper's claims.
- The method produces the same output from inputs of different scales.
- The presentation could be significantly improved. In particular, implementation details are missing, which limits the reproducibility of the paper.

The strengths seem to be limited to introducing a new approach for counting. However, the weaknesses, particularly the lack of experimentation in traditional crowd-counting datasets, are more extensive. The major concerns are that the experiments do not support the claims due to missing comparisons in crowd-counting datasets and with other methods. Thus, I recommend rejecting the paper.

**Additional Comments On Reviewer Discussion:**

Reviewer Jvjv mentions that the presentation of the paper requires major improvement. Moreover, the problem setup is not clear, as the problem assumes that the input image scale is unknown rather than addressing scale variations within an image. Yet, it evaluates on remote sensing datasets where the images often have a uniform scale. The paper is not evaluated on crowd counting datasets and few-shot counting datasets where the scale/invariant issue is relevant. The authors provided comments and replied to the reviewer, particularly by adding results for UCF-QNRF. The reviewer commented that while the clarification was helpful, the paper still requires major edits and complained that the presented results did not include all recent results. Moreover, the claims raised by the authors were not backed up by the results they presented.

Reviewer mauM mentioned that the compared methods are limited and several methods are missing. Similar to Reviewer Jvjv, this reviewer mentioned that the objects have a similar size, which challenges the justification of the method for multi-scale purposes. The reviewer also requested comparisons against traditional dense crowd datasets. While the authors showed the same results on the UCF-QNRF dataset, the reviewer noted that the results are not as performant as the compared methods.

Reviewer dyoE mentioned that the paper is hard to follow and that the results do not support the multi-scale claims. Despite the rating, the comments and concerns are in line with the other reviewers, who have a more negative view of the paper. The authors replied to the reviewer's concerns, but the reviewer did not respond.

Reviewer fHTu raised the issue about the lack of a detailed explanation of the inner workings of the method. Similar to the other reviewers, this one raised questions about missing baselines across various resolutions and asked for additional experiments regarding the robustness of images at extreme resolutions. The authors presented the results as they did for the other reviewers and replied to the comments. However, the reviewer did not reply.

After the rebuttal, I asked the reviewers about their decisions and the divergent scores regarding the comments presented in the discussion, but none replied.

Overall, I see a common thread that the paper lacks experimental results robust enough to validate scale-invariant claims of the proposal and lacks extensive comparisons in traditional databases for crowd counting. The positive reviewers still raised similar issues as the more negative ones, and the strengths are limited. Thus, I recommend rejecting the paper.

---

### Decision · Program_Chairs · 2025-01-22

Reject